# Development of behavioral rules for upstream orientation of fish in confined space

**David C. Gisen**[1]*, **Cornelia Schütz**[2], **Roman B. Weichert**[1]

**1** Waterways and environment unit, Bundesanstalt für Wasserbau, Karlsruhe, Germany, **2** Animal ecology unit, Bundesanstalt für Gewässerkunde, Koblenz, Germany

* david.gisen@baw.de

**Data Availability Statement:** All relevant data are within the manuscript and its Supporting Information files. All model source code files are available from Github, https://github.com/baw-de/ELAM-flume.

## Abstract

Improving the effectiveness of fishways requires a better understanding of fish behavior near hydraulic structures, especially of upstream orientation. One of the most promising approaches to this problem is the use of model behavioral rules. We developed a three-dimensional individual-based model based on observed brown trout (*Salmo trutta fario*) movement in a laboratory flume and tested it against two hydraulically different flume set-ups. We used the model to examine which of five behavioral rule versions would best explain upstream trout orientation. The versions differed in the stimulus for swim angle selection. The baseline stimulus was positive rheotaxis with a random component. It was supplemented by attraction towards either lower velocity magnitude, constant turbulence kinetic energy, increased flow acceleration, or shorter wall distance. We found that the baseline stimulus version already explained large parts of the observed behavior. Mixed results for velocity magnitude, turbulence kinetic energy, and flow acceleration indicated that the brown trout did not orient primarily by means of these flow features. The wall distance version produced significantly improved results, suggesting that wall distance was the dominant orientation stimulus for brown trout in our hydraulic conditions. The absolute root mean square error (RMSE) was small for the best parameter set (RMSE = 9 for setup 1, RMSE = 6 for setup 2). Our best explanation for these results is dominance of the visual sense favored by absence of challenging hydraulic stimuli. We conclude that under similar conditions (moderate flow and visible walls), wall distance could be a relevant stimulus in confined space, particularly for fishway studies and design in IBMs, laboratory, and the field.

## Introduction

Worldwide, fishways play a key role in efforts to restore upstream-directed fish migration at dams. The widely differing effectiveness of existing projects [1, 2] illustrates that their design is still challenging, especially for species other than Pacific salmon [3]. Improving fishway effectiveness for all species requires a better understanding of fish behavior near hydraulic structures, especially of orientation and navigation [4, 5]. One of the most promising approaches to better understand such behavior and "a high research priority" [5] is the development of

**Funding:** This work was funded by the German Federal Ministry of Transport and Digital Infrastructure (BMVI, https://www.bmvi.de). The funders had no role in study design, data collection and analysis, decision to publish, or preparation of the manuscript.

**Competing interests:** The authors have declared that no competing interests exist.

behavioral rules. Behavioral rules are the logical and/or algebraical formulation of behavioral hypotheses. They can be implemented and tested in individual-based models (IBMs, [6]).

IBMs enable one to implement the concepts of orientation (directional response to local conditions, [4]) and navigation (movement towards a goal outside the local sensory environment, [4]) rather accurately. For example, in a number of recent fish movement IBMs, orientation is the outcome of behavioral rules using local stimuli, while navigation emerges from the sum of orientation decisions over time (e.g., [7–11]). Together with their ability to integrate CFD (computational fluid dynamics) model results, IBMs are therefore highly suitable to support fishway design.

A preferred object of research for fishway design are hydraulic stimuli, because they can be manipulated to optimize passage efficiency. The most widely used stimulus in fish orientation IBMs, regardless of whether the model considers upstream or downstream fish migration, is velocity direction [8, 11–14]. It is a fundamental stimulus, as it informs positive and negative rheotaxis, which are crucial for guiding upstream and downstream navigation, respectively [15–17]. Velocity magnitude is also commonly considered [8, 10–14] and especially important for upstream migration, since the energy demand for swimming against the flow is proportional to the cube of relative fish velocity [18].

Apart from velocity direction and magnitude, stimulus selection to formulate behavioral rules remains difficult [8]. To integrate information on turbulence into IBMs, the turbulence kinetic energy (TKE) is a common choice [12, 13]. Turbulence may also be integrated as velocity fluctuation [11]. Other approaches to capture hydraulic information in IBMs include using spatial acceleration [8–10] and the related concepts of spatial velocity gradient and strain rate [7, 13].

Fish studies from laboratory flumes offer further potential stimuli for modelling upstream orientation, for example the horizontal Reynolds shear stress ($\tau_{xy}$), turbulence intensity, and eddy orientation and scale (see [19] for a framework). However, results are manifold and could depend on the specific setup and/or species tested. In different studies, dace preferred adapted turbulence levels (TKE and $\tau_{xy}$) [20], adult Iberian barbels primarily avoided areas of high $\tau_{xy}$ [21], and brown trout preferred low drag areas [22]. Eddy orientation and scale did influence Iberian barbel behavior [21], but are still difficult to define in common vertical-slot fishways [23] and are not used in IBMs to date. As for IBMs, the selection of hydraulic stimuli for laboratory studies remains controversial [22].

Non-hydraulic external stimuli for fish orientation, such as visual, olfactory, acoustic, thermal, and magnetic stimuli, are rarely investigated in flows that are comparable to a fishway flow and the authors are not aware of corresponding fish orientation IBMs that include such stimuli. This is particularly notable for vision, as it is considered important for orientation [24, 25]. Published laboratory and field studies on upstream orientation using visual stimuli as explanatory variable are limited to general phototaxis, i.e. attraction or repulsion [16, 26].

Basic motivation to swim against the flow is crucial for successful upstream migration. However, such motivation may vary widely between species and even between individuals [3]. Existing upstream orientation IBMs avoid modeling motivation as an internal state by determining that fish constantly try to move upstream [9, 11–14]. However, this approach prevents the potentially important ability to drift, recover, and repeat passage attempts after initial failure [27]. Modeling variable internal states to elicit differing behavioral responses to identical situations could be one way to approach this problem.

Furthermore, existing IBMs for upstream migration still await quantitative validation. They are either tested qualitatively, e.g. using visual track comparisons, or untested due to a lack of observation data. This is a remarkable contrast to the comprehensive quantitative testing available for downstream migration IBMs [7, 8, 10].

In the present study, we thus developed a new IBM for fish upstream orientation and navigation in confined space to support fishway design. It was tested against detailed data of brown trout movement in a laboratory flume, and is the first such model enabling holding behavior, recovery, and thus repeated passage attempts. Five external stimuli were selected for testing based on the literature:

- positive rheotaxis as a fundamental behavior in upstream migrating fish,

- velocity magnitude as being fundamentally related to energy demand,

- TKE as one of the most common and general turbulence variables,

- spatial acceleration as having been tested against the largest data sets by far,

- wall distance as a potential proxy for visual orientation.

The present paper describes how we (1) defined movement patterns from brown trout flume data, (2) developed and tested a CFD model to generate stimuli data, (3) developed and tested a behavioral model to reproduce the movement patterns, and (4) examined which of the five external stimuli would best explain upstream brown trout orientation and navigation in confined space.

## Methods

### Flume setups

For developing and testing the behavioral model, we used unpublished brown trout movement data from a previous, unrelated study (wild *Salmo trutta fario*, n = 66, mean body length ± standard deviation (SD) = 0.27±0.04 m, Schütz et al. [28]). Fish were caught in a close-by river during their spawning period in October/November, when an increased motivation to migrate can be expected [28]. The study was conducted in a wide indoor flume (Fig 1). The external walls and bottom consisted of smooth acryl glass and metal and the internal walls consisted of coated wood. Water temperature was 16.2–18.2°C. Water was clear and evenly illuminated. All care and procedures involving handling and holding fish were conducted as stated and permitted by the district government Karlsruhe (license AZ 35–9185.82/A-6/16).

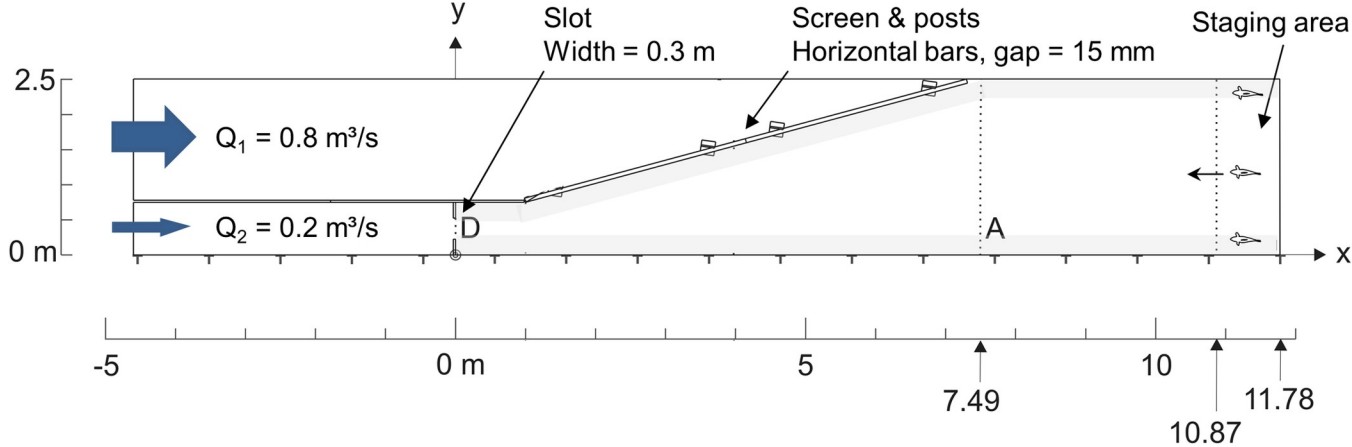

**Fig 1. Plan view of flume setup 1.** Total length and width = 16.38 m x 2.50 m, water depth = 0.60 m. Flow direction was from left to right. The slot was removed in setup 2 to alter the flow field. Transparent gray areas mark left and right zones (in flow direction) for pattern P1 (defined below, section "Movement patterns"). *A* and *D* mark control lines to filter inactive fish and to define pattern P5.

We used two hydraulically different setups from Schütz et al. [28] to test the application range of our model. Setup 1 included a jet created by a slot; setup 2 had no slot and consequently, no jet. The total discharge of $Q = 1.00$ m³/s (used in both setups) produced a jet velocity of $U = 1.5$ m/s, which is typical for slots of large multi-species fishways in Germany. The flume was wide enough to enable distinct lateral movement, but narrow enough to influence behavior through the walls and screen.

Fish were released unmarked in batches of three at the downstream end (staging area in Fig 1) and were used only once to avoid learning effects. Fish positions were manually recorded in real time independently by two biologists upon notable change. A change of position was defined as either:

- crossing a meter line of the flume in longitudinal direction

- moving between three zones in lateral direction: close to left wall, close to right wall, far from wall (close meaning max. ~ 0.25 m)

- moving between three zones in vertical direction (close to bottom, close to surface, in between (close meaning max. ~ 0.15 m)

Moreover, it was recorded if fish swam alone or in groups of two or three. Records were attributed to the current test minute. During postprocessing, multiple position changes within the same minute were distributed evenly over the seconds (S1 and S2 Appendices). Data were verified qualitatively using video records.

To ensure that all fish were actually motivated to move during the experiment, only fish that crossed line *A* (Fig 1) within 30 min after removing the staging mesh were considered valid. A trial was finished either after all three fish passed line *D* or 60 min after the first crossing of line *A*. Data points following a passage of line *D* were excluded. After filtering, we obtained $n = 25$ tracks for setup 1 and $n = 24$ tracks for setup 2.

## Movement patterns

We processed the trout track data using *patterns* in the sense of pattern-oriented modeling [29] using MATLAB R2018b. We defined five patterns, P1-P5, to capture the most striking spatial behaviors observed during the original experiment. They included a preference for wall and bottom proximity as well as frequently swimming back and forth (turns):

**P1, lateral distribution.** The flume was divided into a left, middle, and right zone (looking downstream, Fig 1). The dividing lines were located at a lateral distance of $\Delta y = 0.25$ m (10 % flume width) to one of either side walls or the screen. Time spent by single fish (i.e., center of mass) was summed for each zone. To account for the differences in total track duration [see 30], resulting time sums per fish and zone, $t_{zone,i}$, were divided by the particular fish's track duration, $t_{s,i}$, to get relative track fractions, $s_{zone}$. Finally, $s_{zone}$ was averaged by the fish count, $n$, to get:

$$\overline{s}_{zone} = \frac{1}{n}\sum_{i=1}^{n} \frac{t_{zone,i}}{t_{s,i}} * 100 \ [\%] \qquad (1)$$

where $i$ was a fish iterator. The resulting three lateral zone averages constituted pattern P1. As it is the most direct outcome of upstream orientation, matching it was a major goal of model development.

**P2, vertical distribution.** Vertically, the flume was divided into a surface, middle, and bottom zone. The dividing lines were located at a vertical distance of $\Delta z = 0.15$ m (25 % water depth) to either the water surface or flume bottom, respectively. Tracks were averaged using Eq 1 to obtain three vertical zone average values. Due to the shallow water, we expected P2 to be of less significance for orientation.

**P3, turn zone and P4, few turns.**   Turns were defined as changes in the longitudinal movement direction, $x$, leading to a displacement $\Delta x \geq 2$ body lengths (BL) before the next direction change. About half of the fish in both data sets performed 4 or less turns. To avoid bias, the data set was divided into turning ($> 4$ turns, P3, $n_{turns}$) and few-/no-turning fish ($\leq 4$ turns, P4, $n_{few\text{-}turns}$). For computing P3, the flume was divided longitudinally into three zones of equal length, in which the relative turn shares were computed and averaged by $n_{turns}$ analog to Eq 1. P4 was defined as $n_{few\text{-}turns}/n$.

**P5, arrival rate.**   The count of fish navigating the total length of the experimental flume area and crossing line $D$, divided by the count of fish crossing line $A$, formed P5: $n_D/n \cdot 100$ [%].

For use in discussion, we also calculated the lateral distribution of fish at the start position and the proportion of track time fish spent without other fish nearby (i.e., not shoaling).

## Computational fluid dynamics model

A CFD model was employed to calculate the velocity, TKE, and acceleration fields of both set-ups. We applied the free-surface solver *interFoam* of OpenFOAM 2.3.1 [31] to solve the 3D incompressible URANS (unsteady Reynolds-averaged Navier-Stokes) equations.

A hexahedron-dominant unstructured mesh was generated using *snappyHexMesh*. Base resolution was uniform $\Delta_{xyz} = 5$ cm, with local refinement to $\Delta_{xyz} = 1.25$ cm around the slot and screen posts (Fig 1). Horizontal screen bars were omitted, as their influence on the velocity and turbulence field was negligible with respect to behavioral data accuracy [32]. Total cell count for setup 1 was 848,694.

Inlet flow rates and outlet fixed water level were set to the laboratory values. For turbulence closure, we chose the $k$-$\omega$-SST model. At the inlet, TKE ($\triangleq k$) was set to $k = 0.001$ m$^2$/s$^2$ and the specific rate of TKE dissipation was set to $\omega = 1$ Hz. No-slip conditions (velocity vector $U = (0,0,0)$) and a small sand-equivalent roughness coefficient ($k_S = 1 \cdot 10^{-5}$ m) were set at the walls and bottom.

The fields of velocity $U$ and water/air distribution $\alpha_{water}$ were initialized with a largely converged solution and ran for 80 s of simulated time. $U$ was averaged over the final 20 s to obtain the steady flow velocity vector field $\overrightarrow{U}_m$. $\overrightarrow{U}_m$ can be expressed either as three-component vector $(u,v,w)$ or as magnitude $U_m$ along with its horizontal and vertical angles $\gamma_m$ and $\beta_m$.

TKE is defined as half the sum of the velocity components' variances (squared turbulence intensities):

$$TKE = \frac{1}{2}\overline{U_i' U_i'} = \frac{1}{2}\left(\overline{u'^2} + \overline{v'^2} + \overline{w'^2}\right) \left[\frac{m^2}{s^2} = \frac{J}{kg}\right]$$ (2)

using Reynolds decomposition, $U_i = \overline{U_i} + U_i'$.

Spatial acceleration $a$ is caused by changes in the flume cross-section and by friction, in contrast to temporal acceleration caused by transient processes. Spatial acceleration is the product of velocity and velocity gradient (which comprises strain rate and rotation rate). In 3D Cartesian coordinates, it reads

$$a = \begin{pmatrix} a_x \\ a_y \\ a_z \end{pmatrix} = \left(U_m^T \, grad \, U_m\right)^T = \left((u \quad v \quad w) \begin{pmatrix} \frac{\partial u}{\partial x} & \frac{\partial v}{\partial x} & \frac{\partial w}{\partial x} \\ \frac{\partial u}{\partial y} & \frac{\partial v}{\partial y} & \frac{\partial w}{\partial y} \\ \frac{\partial u}{\partial z} & \frac{\partial v}{\partial z} & \frac{\partial w}{\partial z} \end{pmatrix}\right)^T \left[\frac{m}{s^2}\right]$$ (3)

where the superscript $^T$ denotes transposition from column to row vector. Its magnitude $|a|$, $U_m$, and TKE were used as hydraulic stimuli variables in the behavioral model.

## Behavioral model description

A complete, detailed model description, following the ODD (overview, design concepts, details) protocol [33, 34] can be found in S4 Appendix. Here, we provide a summary. The overall purpose of our model was to find behavioral rules for fish that could enable predictions of fishway attraction and passage efficiency. Specifically, we addressed the following question: How well do different external stimuli explain orientation and navigation of upstream moving brown trout?

The entities of the model were fish and mesh cells. Fish were represented as mesh-independent, spatially explicit points in 3D space. Their key state variables were position, motivation, and fatigue. Fish movement was modeled kinematically, i.e. there was no influence of fish on the flow field. Key state variables of the mesh cells were the cell label number and the hydraulic variables $|a|$, $U_m$, TKE, and $\alpha_{water}$. Spatially, the behavioral model covered the fish-accessible areas of the CFD model (i.e. without the air phase and without areas upstream of the screen). The temporal extent was 60 min of clock time, equal to the maximum laboratory trial duration, divided into constant time steps of $\Delta t = 0.5$ s.

Each model run used $n_{model} = 100$ fish of BL = 0.27 m (as measured), each being assigned to either a "fast" or "slow" category. In this way we tried to model individual traits which could influence migratory tendencies, such as being bold or shy [35] or being physically strong or weak. Model fish were positioned only in the left and right zones of the staging area (Fig 1) as per observations. Their exact distribution was controlled by a parameter.

Every time step, the model computed a new position for every fish. First, a sensory ovoid consisting of one point in the fish center and six surrounding points was determined [7]. The surrounding points were placed at an imaginary skin/water interface, approximately where real fish detect water motions by their lateral line system [36]. Hydraulic variables were interpolated to these points. Wall distances were computed towards the front and sides.

Next, we aimed to summarize all fish characteristics possibly acting on longitudinal movement by two opposing variables: motivation (to swim upstream), $M$, and fatigue, $F$ (both range 0.0–1.0). The variables resembled the typical internal states "need" and "cost" [4]. Their balance determined the following movement decision. We chose this rather basic representation of internal stimuli because our focus was on orientation and external stimuli. Still, it enabled model fish to drift, recover, and swim upstream again, a behavior which we consider crucial for modeling fish migration through fishways [27]. Its success was measured foremost by patterns P3 and P4.

$M$ was increased proportionally with time when the model fish made no spatial progress, presuming there was a principle motivation to move upstream. The maximum increase rate was reached after $k_{M,f} = 20$ s for "fast" fish and $k_{M,s} = 130$ s for "slow" fish (default values, varied later). $F$ was computed proportionally to local flow velocity and swim speed. It reached the maximum value at a burst speed of $k_F = 25$ BL/s [37]. We note that more realistic speed-fatigue relations are available [37, 38], but they were not required for our purpose of modeling orientation. From the balance of time-averaged $M$ and $F$, one of three horizontal behavioral rules was selected (Fig 2):

- migrating (moving upstream),

- holding (maintaining position),

- drifting (moving downstream with limited random deviation from the flow vector).

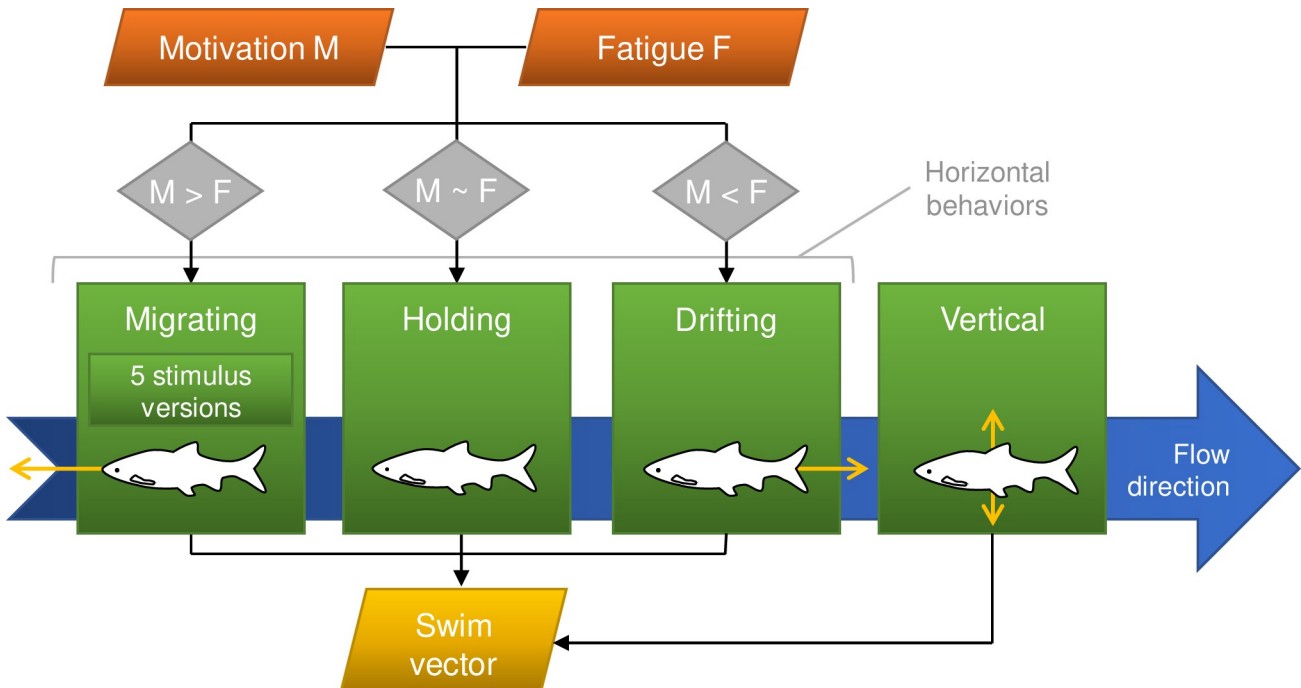

**Fig 2. The process of computing the swim vector.** Based on the balance between motivation and fatigue, a horizontal behavioral rule is selected. *Migrating* is the key behavioral rule for which five versions are contrasted. *Holding* means maintaining the position. *Drifting* includes swimming against, but slower than the flow. In combination with the vertical behavioral rule, the final swim vector is determined.

*Migrating* was the key behavioral rule. In its baseline version it only depended on the velocity direction. It set the horizontal and vertical swim angles against the velocity direction, as we judged rheotaxis to be a fundamental behavior in fish that is always active (e.g. in Mexican tetra [16]). To also capture smaller, unpredictable behavior variations, a limited random angle was added to or subtracted from the horizontal swim angle. Migrating was expected to have the most direct effect on pattern P1.

Our guiding question ("how well do external stimuli explain upstream orientation?") was addressed by adding four different stimuli for orientation to the baseline migrating behavioral rule. That is, the horizontal swim angle could be adjusted in either one of five alternative stimulus versions:

- Baseline (rheotaxis): Against velocity direction & towards random side by a limited random angle as described above.

- Velocity: Baseline & towards side of lower velocity magnitude. *Rationale*: Avoiding zones of high velocity to save energy (low-velocity seeking) is a common strategy for fish and was observed e.g. in laboratory flumes for longnose dace [39] and brown trout [22] and modeled in an IBM for carp and sturgeon [11].

- TKE: Baseline & towards front or side with the smallest TKE difference towards the fish center (constant turbulence level). *Rationale*: As it is still unclear how turbulence affects orientation in a specific situation or for a single species and fish size, we chose one of the most common and general 3D turbulence variables, containing the turbulence intensity. Orientation along a constant, adapted turbulence level was observed before in dace [20] and modeled in an IBM [12].

- Spatial acceleration: Baseline & towards side of higher |a| magnitude. *Rationale*: Acceleration is the hydraulic stimulus tested against the largest data sets by far, although for downstream migration [8, 10]. As it also contains the strain rate [13] within the velocity gradient tensor, it is an information-rich representation of the flow field and we consider it potentially useful also for upstream migration [e.g., 9]. We chose attraction towards higher acceleration to mimic our laboratory results.

- Wall distance: Baseline & towards side of shorter wall distance. *Rationale*: Within the confined space of our model, wall distance can serve as a potential proxy for the little investigated, but potentially important [24, 25] visual orientation. Again, we chose attraction towards shorter wall distance to mimic our laboratory observations.

An independent vertical behavioral rule was always active. It limited vertical movement using an elevation difference [7, 8] and was primarily gauged by pattern P2.

The output of the process described was a 3D swim vector for each fish and time step. The new position was determined from this swim vector, the flow vector at the fish's position, and the time step width. Overall navigational success was measured by pattern P5.

**Code and speed.** The code of the behavior model and its software framework can be found in in S5 Appendix. The software framework was based on the Fortran 90 code for downstream migrating smolts used in Goodwin et al. [8]. It accounted for computer simulation dynamics such as variable storage, sensory point creation, vector transformation from/to Cartesian coordinates, and pseudo-random number creation. We modified it to work with unstructured, polyhedral meshes in OpenFOAM 4.1 using C++. Computation time varied depending on how fast fish exited the domain. A typical runtime for a single model run with $n_{model}$ = 100 fish was about 85 s, using 8.1 GB RAM for up to 7200 time steps on one core of an Intel E5-2660-v3 ("Haswell") CPU.

## Model function test

Besides behavioral rules, parameters (i.e., variables which do not change during a model run, but can change between model runs) are the second main component of a behavioral model. Our methods for testing their influence are described in this section before returning to our actual purpose of testing different stimuli in section "Stimulus version test".

**Evaluation metric RMSE.** To rate the quality of each model run, we computed pattern values P1-P5 in analogy to the laboratory patterns from $n_{model}$ = 100 fish. As an evaluation metric, we computed a weighted root-mean-square error (RMSE) between all model and laboratory pattern mean values (Table 1 in Results) for each model run as

$$RMSE_{model-run} = \sqrt{a_1 \Delta P_1 + a_2 \Delta P_2 + a_3 \Delta P_3 + a_4 \Delta P_4 + a_5 \Delta P_5} \tag{4}$$

$$\text{Average of pattern P1-P3:} \quad \Delta P_i = \frac{1}{3} \sum_j (P_{i,j,model} - P_{i,j,lab})^2 \tag{5}$$

$$\text{Pattern P4-P5:} \quad \Delta P_i = (P_{i,model} - P_{i,lab})^2 \tag{6}$$

where *a* was a pattern weight factor chosen per pattern importance ($a_1$ = 0.3, $a_2$ = 0.1, $a_{3-5}$ = 0.2), *i* = 1..5 was a pattern iterator, and *j* = 1..3 was a pattern value iterator. As P1-P3 consisted of three pattern values each, these were averaged. A value of RMSE = 0 implies perfect agreement between model and laboratory results. We chose RMSE as it is both simple and penalizes large differences, which makes it useful for filtering outliers and achieving an overall good agreement.

The different weights of patterns P1 and P2 reflected their expected importance for modeling orientation (section "Movement patterns"). We also tested equal weighting per pattern average value and equal weighting per pattern value, but obtained negligible effects on the stimulus version ranking. We focused on comparing the mean instead of SD values to reduce the complexity of model development, accepting that SD agreement cannot be evaluated.

**Random seed sensitivity.** The behavioral rules used pseudo-random numbers which were generated from a random seed number fixed for each model run. For identical seeds, results were identical. To minimize and to quantify influence of the random seed choice, we tested each parameter set with $N_r$ = 10 different seeds. Then, we averaged the resulting $RMSE_{model-run}$ values to obtain one RMSE value per parameter set. As seeds, we used ten true random numbers from www.random.org in the range from 1–999: 656, 36, 849, 934, 679, 758, 743, 392, 655, 171.

**Parameter set generation.** In total, the model had 7 fixed and $D$ = 20 variable parameters (Table 2 in Results and Table 3 in S4 Appendix, p. 8). The latter could be split into $D_1$ = 14 parameters for general movement, $D_2$ = 3 parameters for initialization, and $D_3$ = 3 individual parameters that were used only for single stimulus versions.

For parameter sensitivity testing (next section, "Parameter sensitivity test") and stimulus version testing (section "Stimulus version test"), we needed to generate a number of parameter sets with systematic variation in their values. The values of an initial default parameter set were established by trial-and-error. For variation, we chose another 5 values in equal distances around each default value to test a total of $p$ = 6 values for each parameter. The default values were either at the 3rd or 4th place of the range (Table 3 in S4 Appendix, p. 8).

The full parameter space would comprise $p^D \approx 3.7 \cdot 10^{15}$ parameter sets. We sampled it using the revised Morris method ([40], $r$ = 1000 random trajectories, $T$ = 50 optimum trajectories) implemented in Python 3.7.4 with SALib 1.3.8. We obtained $n_{sets,base}$ = $(D+1) \cdot T$ = 1050 systematically varying base parameter sets (Fig 3, box 1). They consisted of $D$ parameter values each and each set differed from the next one by exactly one parameter value. Sets differing only in a stimulus-dependent parameter value ($D_3$ parameters) were used only for the corresponding stimulus version.

**Parameter sensitivity test.** To identify parameters of strong and negligible influence, we analyzed parameter sensitivity using the wall distance stimulus version (Fig 3, box 2). As the acceleration/TKE stimulus threshold parameter was not required for this version, the number of parameter sets decreased by $T$ variations to $n_{sets,sens}$ = 1000 parameter sets. As described in section "Random seed sensitivity", for each parameter set we ran the model with $N_r$ random seeds and averaged the resulting $RMSE_{model-run}$ values to obtain one RMSE value per parameter set (Fig 3, box "Random seed sensitivity").

Results were analyzed by means of revised Morris screening [40, 41]. This method uses the effect of single parameter variations on the RMSE value to determine an influence measure, $\mu^*$, and an interaction measure, $\sigma_M$, for each parameter. We used $\mu^*$ for ranking. For further interpretation, the values of the measures needed to be classified as "high" or "low". This is usually achieved graphically [e.g., 40]. We took a more quantitative approach through normalizing $\mu^*$ and $\sigma_M$ by their respective maximum values and defining (arbitrary) thresholds for "high" values at $\mu^*/\mu^*_{max} > 0.6$ and for "low" values at $\mu^*/\mu^*_{max} \leq 0.1$ in dependence of the results (section "Parameter sensitivity"). We neglected three parameters with "low" influence in the subsequent stimulus version tests.

## Stimulus version test

To address our guiding question ("How well do different stimuli explain upstream orientation?"), we compared the five stimulus versions of the "migrating" behavioral rule in both

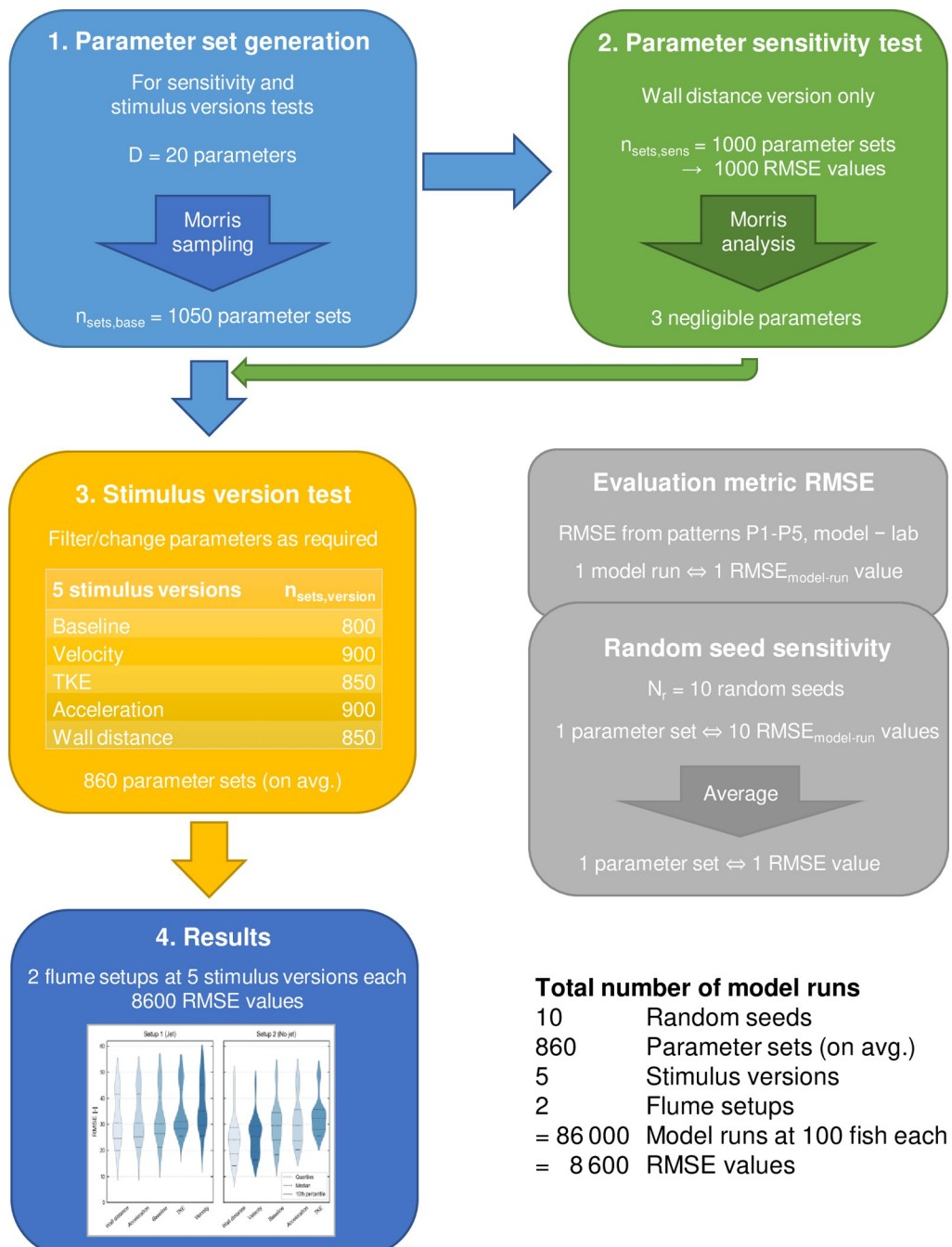

**Fig 3. The process of testing our behavioral model.** The process starts with the generation of varying parameter sets, which are used for testing parameter sensitivity and contrasting stimulus versions in the following. Each box corresponds to one text section.

flume setups (Fig 3, box 3). All other behavioral rules of the model (including behavior selection, holding, drifting, and vertical behavior) were kept constant and were only tested implicitly (for a protocol, see [29]).

**Parameter influence.** Model parameters required special attention for the stimulus version test due to their large influence on the RMSE result. To eliminate the risk of favoring a

stimulus by a certain parameter set, we applied a large number of parameter sets to each of the stimulus versions. We re-used the varying base parameter sets generated for parameter sensitivity testing (section "Parameter set generation"), but removed the three negligible parameters identified in sensitivity analysis. Using the same parameter sets for all versions ensured comparability. Only minor differences from the individual parameters (that were used only for single stimulus versions) were inevitable:

- the threshold factor was used only by the velocity and acceleration versions;

- the default swim angle for the wall distance stimulus was half the default swim angle of the velocity, TKE, and acceleration versions. It was not used in the baseline version.

These differences were also reflected in the slightly varying number of parameter sets tested per stimulus version, $n_{sets}$. The average was $n_{sets,avg}$ = 860 parameter sets in a range of 800–900 (Fig 3, box 3).

**Running.**   For each parameter set, we repeatedly ran the model with $N_r$ random seeds and averaged the resulting $RMSE_{model-run}$ values to obtain one RMSE value (section "Random seed sensitivity"). From the combination of parameter sets ($n_{sets,avg}$ = 860), flume setups ($n_{setups}$ = 2), and stimulus versions ($n_{stimuli}$ = 5), we received a total of 8600 RMSE values (Fig 3, box 4).

**Ranking and analysis.**   Since sampling of the parameters was optimized to cover the parameter space (section "Parameter set generation") and not towards better RMSE values, the tests produced many irrelevant, high RMSE values which masked differences between the stimulus versions. We ranked versions by the 10[th] percentile of RMSE values to neglect the irrelevant values and to accent relevant low (optimum) values instead.

To evaluate if differences between the stimulus versions were statistically significant ($p < \alpha$ = 0.01), we computed the *p*-values between their RMSE values. We only considered values below the 10[th] percentile to cut off irrelevant RMSE values while preserving enough RMSE values for a statistically valid comparison. We used SciPy 1.0.1 with scikit-posthocs 0.6.6 to perform a Kruskal-Wallis test for all versions of a setup and a post-hoc Nemenyi test for pairwise tests between the wall distance version and the remaining versions.

During both ranking and statistical analysis, we treated the setups separately, as setup 1 was more demanding on the model due to the more heterogeneous flow field and the slot geometry.

Finally, to facilitate vivid understanding and discussion, we classified the versions qualitatively. The best version per setup was classified as "good". Versions differing significantly ($p < \alpha$ = 0.01) from the best version were classified as either "moderate" or "poor", depending on their 10[th] percentile value. Versions not differing significantly were classified as "good".

## Results

### Movement patterns

Analysis of the movement patterns provided several insights into observed trout behavior (Table 1). Pattern P1 indicated that brown trout spent the major part of trial time close to the lateral walls and screen, i.e. in a rather small share of the flume width. P2 showed that fish swam close to the bottom almost exclusively. P3-P5 were less distinct. Large SD values indicated wide individual differences in observed behavior for P1 and P3.

Brown trout start positions were distributed to the P1 left/middle/right zones 68 % - 4 % - 28 % (setup 1) and 46 % - 13 % - 42 % (setup 2). They spent an average of about 60 % of track duration without other fish nearby.

**Table 1. Mean and standard deviation (SD) of brown trout patterns in the two flume setups.**

| Pattern | Flume zone | | | Setup 1 (jet) | | | Setup 2 (no jet) | | |
|---|---|---|---|---|---|---|---|---|---|
| | Name | Extent | $n$ | mean | SD | $n$ | mean | SD |
| | (facing downstream) | (m) | | (%) | (%) | | (%) | (%) |
| P1 Lateral distribution | Left | 0.25 | 25 | 33 | ±37 | 24 | 38 | ±39 |
| | Middle | 2.00 | 25 | 10 | ±12 | 24 | 6 | ±10 |
| | Right | 0.25 | 25 | 57 | ±38 | 24 | 56 | ±40 |
| P2 Vertical distribution | Surface | 0.15 | 25 | 1 | ±2 | 24 | 4 | ±17 |
| | Middle | 0.30 | 25 | 1 | ±2 | 24 | 0 | ±1 |
| | Bottom | 0.15 | 25 | 99 | ±3 | 24 | 96 | ±17 |
| P3 Turn zone | Upstream | 3.25 | 9 | 24 | ±19 | 12 | 27 | ±17 |
| | Middle | 3.25 | 9 | 29 | ±17 | 12 | 40 | ±16 |
| | Downstream | 3.25 | 9 | 47 | ±16 | 12 | 33 | ±26 |
| P4 Few turns | - | - | 15 | 60 | - | 12 | 50 | - |
| P5 $n_D/n$ | - | - | 25 | 84 | - | 24 | 79 | - |

Values rounded to the closest integer. $n_D$ and $n$, number of fish at lines $D$ and $A$.

## Computational fluid dynamics model

CFD model results were in good agreement with acoustic Doppler velocimeter measurements from the laboratory [32]. Advection dominated the flow field; boundary roughness influence was limited to a few centimeters in both setups (Fig 4, S3 Appendix). Absence of the jet in setup 2 considerably reduced mean velocity $U_m$, acceleration magnitude, and TKE on the right-hand flume side (facing downstream). The flow field of both setups was asymmetric in lateral direction. The bulk flow velocity was $U_{bulk}$ = Q/(width*water depth) ≈ 0.67 m/s. In large parts of the jet region of setup 1, flow velocity was about $U_m$ ≈ 0.8 m/s.

## Random seed and time step sensitivity

To evaluate the influence of the different random seeds used, we compared the median coefficient of variation ($\sigma/\mu$) for all ten setup/stimulus combinations. The values were low and ranged from 0.02–0.04, suggesting that our results were independent of the random seed chosen.

Halving the default time step to $\Delta t$ = 0.25 s for the best wall distance parameter set deteriorated the RMSE in setup 1 and improved it in setup 2. Doubling the time step deteriorated the RMSE in both setups.

## Parameter sensitivity

Parameter sensitivity was computed for the wall distance stimulus version to identify parameters of strong and negligible influence. Setup 1 exhibited seven parameters of strong influence ($\mu^*/\mu^*_{max}$ > 0.6, Table 2), while setup 2 exhibited four parameters of strong influence. The lowest ranks (17–19 in Table 2) comprised three parameters of negligible influence in both setups ($\mu^*/\mu^*_{max}$ ≤ 0.1). There was no first-order (independent) parameter (indicated by a high $\mu^*$ and low $\sigma_M$ value).

Further analysis showed that both swim angles included in the migrating behavioral rule had a high influence and interaction in both setups. This result confirms that modeling orientation was essential for agreement with the laboratory observations. The vertical behavior correction angle was distinctly more important in setup 1 than in setup 2. This was likely caused by vertical currents induced by the head drop at the slot in setup 1.

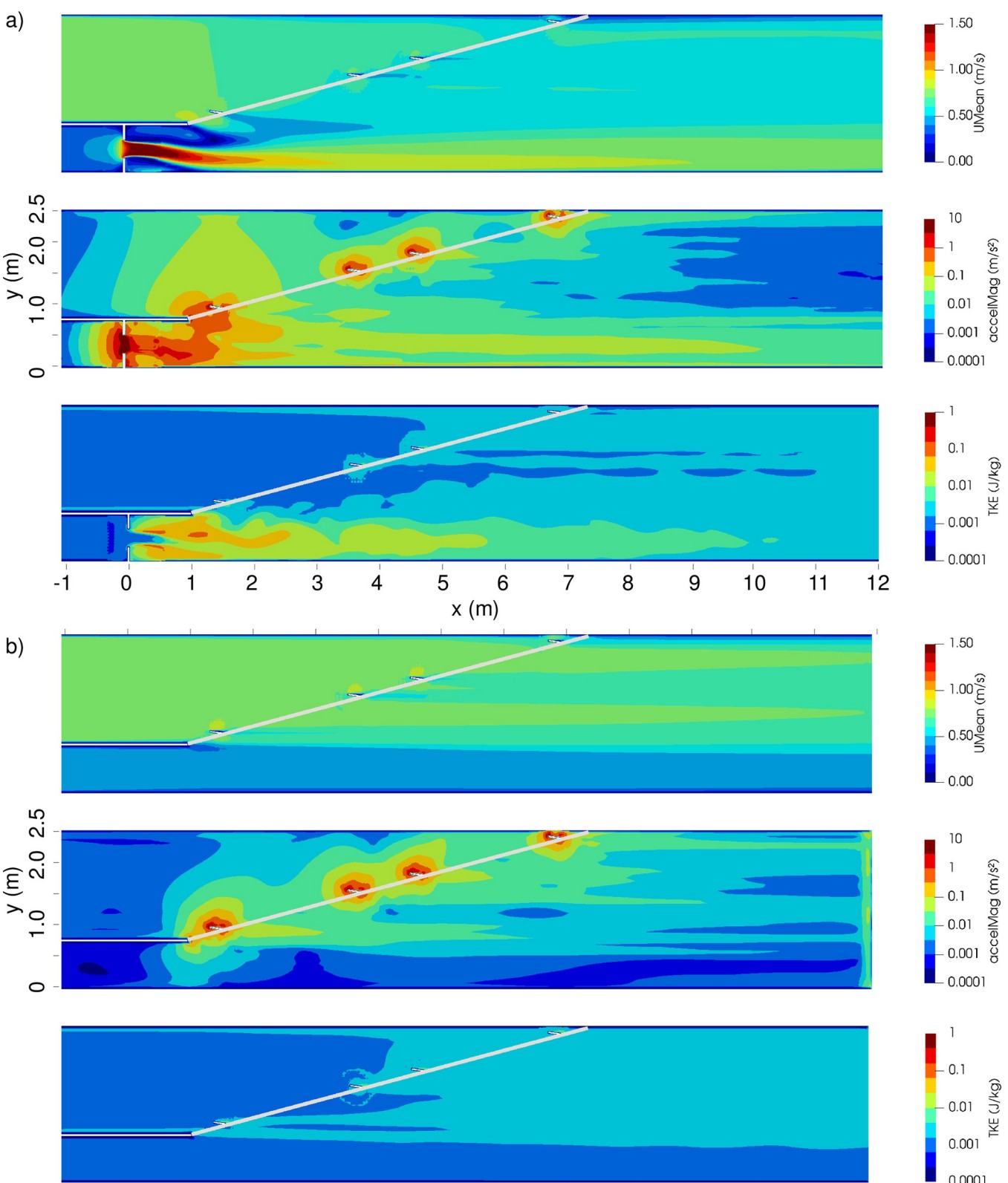

**Fig 4.** Horizontal flow field slices of setup 1 (a) and setup 2 (b) illustrate the effect of the slot and jet and its removal. Flow from left to right. Plane at $z = 0.07$ m above bottom where trout preferred to stay. Water depth = 0.60 m. Sub-panels in (a) and (b) show velocity magnitude $U_m$ (top panels), acceleration magnitude (middle panels), and TKE (bottom panels). Note logarithmic scale for acceleration and TKE.

**Table 2. Parameters of the behavioral model with wall distance stimulus, ranked by normalized influence, $\mu^*/\mu^*_{max}$, in setup 1.**

| Setup 1 (Jet) | | | Parameter name | Setup 2 (No jet) | | | No. |
|---|---|---|---|---|---|---|---|
| Rank | $\mu^*/\mu^*_{max}$ | $\sigma_M/\sigma_{M,max}$ | | Rank | $\mu^*/\mu^*_{max}$ | $\sigma_M/\sigma_{M,max}$ | |
| 1 | 1.00 | 1.00 | Migrating behavior max. random angle | 2 | 0.74 | 0.60 | 14 |
| 2 | 0.82 | 0.87 | Holding behavior extent | 5 | 0.55 | 0.56 | 9 |
| 3 | 0.77 | 0.89 | Migrating behavior wall distance angle | 4 | 0.63 | 0.59 | 16 |
| 4 | 0.70 | 0.86 | Fatigue (decreasing) memory coefficient | 1 | 1.00 | 1.00 | 6 |
| 5 | 0.68 | 0.72 | Vertical behavior correction angle | 16 | 0.12 | 0.21 | 12 |
| 6 | 0.67 | 0.68 | Spot memory coefficient | 3 | 0.68 | 0.89 | 5 |
| 7 | 0.66 | 0.79 | Motivation denominator ("fast fish") | 7 | 0.47 | 0.69 | 3 |
| 8 | 0.56 | 0.64 | Motivation memory coefficient | 6 | 0.47 | 0.60 | 2 |
| 9 | 0.47 | 0.53 | Stuck time threshold | 10 | 0.22 | 0.34 | 13 |
| 10 | 0.41 | 0.46 | Vertical behavior elevation threshold | 13 | 0.13 | 0.17 | 11 |
| 11 | 0.33 | 0.41 | Fatigue denominator | 14 | 0.13 | 0.19 | 8 |
| 12 | 0.32 | 0.37 | Motivation denominator ("slow fish") | 9 | 0.27 | 0.37 | 4 |
| 13 | 0.24 | 0.34 | Motivation initial value | 11 | 0.21 | 0.30 | 1 |
| 14 | 0.24 | 0.29 | Initialization "slow fish right"/"right fish" | 15 | 0.13 | 0.17 | 20 |
| 15 | 0.21 | 0.20 | Initialization "slow fish left"/"left fish" | 12 | 0.13 | 0.34 | 19 |
| 16 | 0.20 | 0.23 | Initialization "left fish"/"total fish" | 8 | 0.44 | 0.44 | 18 |
| 17 | 0.09 | 0.10 | Drifting behavior straight drift probability | 17 | 0.10 | 0.23 | 10 |
| 18 | 0.07 | 0.07 | Fatigue (increasing) memory coefficient | 18 | 0.05 | 0.08 | 7 |
| 19 | 0.04 | 0.04 | Migrating behavior wall detection range | 19 | 0.02 | 0.02 | 15 |

$\sigma_M/\sigma_{M,max}$ is normalized interaction with other parameters. The threshold for "low" values is indicated by a horizontal line. The last column, *No.*, is for reference to Table 3 in S4 Appendix, p. 8.

Parameter sets differing only in one of the three negligible parameters (Table 2, lowest ranks) were removed from the final stimulus version tests to reduce complexity. The sensitivity test was not repeated for stimulus versions other than wall distance as (a) the parameter "wall detection range" was not used in other versions, and (b) the other two negligible parameters were not related to orientation, i.e. were expected to not interact strongly with the chosen stimulus.

## Stimulus versions

We contrasted five stimulus versions for upstream orientation (swim angle selection) in the model and ranked them by their RMSE 10th percentile (Table 3, Fig 5). The Kruskal-Wallis group test obtained $p = 16.2\text{E-}54$ for setup 1 and $p = 1.1\text{E-}78$ for setup 2, suggesting that the stimulus version did influence agreement with the laboratory observations significantly.

Wall distance was the best stimulus version in both setups. Acceleration and velocity differed widely in their ranks between setups, while baseline and TKE did not. In setup 1, the order of acceleration and baseline was not distinct, as well as the order of TKE and velocity (Fig 5). This is reflected in our qualitative rating used for discussion (Table 3). The estimated distributions of the RMSE values are mainly multi-modal due to unfavorable parameter sets (section "Ranking and analysis"), which was a major reason to rank them by their 10th percentile. The overall better results in setup 2 indicate that it was less demanding on the model than setup 1.

The wall distance version with its best parameter set reproduced all five patterns with high accuracy (RMSE = 9 for setup 1, RMSE = 6 for setup 2) (Figs 5 and 6). The finding shows that our model is able to reproduce the system characteristics that are captured by our patterns [6].

**Table 3. Stimulus versions ranked by their RMSE 10th percentile, per setup.**

| Setup | Stimulus version | RMSE 10th percentile | p-Value | Rating |
|---|---|---|---|---|
| 1—Jet | Wall distance | 20 | - | good |
| | Acceleration | 21 | 1.2E-01 | good |
| | Baseline | 21 | 4.0E-01 | good |
| | TKE | 25 | *4.1E-30 | poor |
| | Velocity | 26 | *2.7E-27 | poor |
| 2—No jet | Wall distance | 14 | - | good |
| | Velocity | 16 | 8.9E-02 | good |
| | Baseline | 18 | *1.2E-12 | moderate |
| | Acceleration | 20 | *9.5E-29 | moderate |
| | TKE | 26 | *2.0E-60 | poor |

The *p*-values were calculated using a post-hoc Nemenyi test between wall distance and the particular stimulus version. *p*-values are rounded to two significant digits, percentile values are rounded to integer. Significant differences at $p < \alpha = 1.0E-02$ are denoted by an asterisk (*) and cause the rating to be either "moderate" or "poor". Underlying data in S6 Appendix.

## Discussion

### Stimuli for orientation

In this work, we combined the very different fields of ethology, hydraulic engineering, and behavior modeling to approach a classical problem of behavioral ecology: How can orientation behavior be explained through external stimuli? The resulting movement patterns, flow fields,

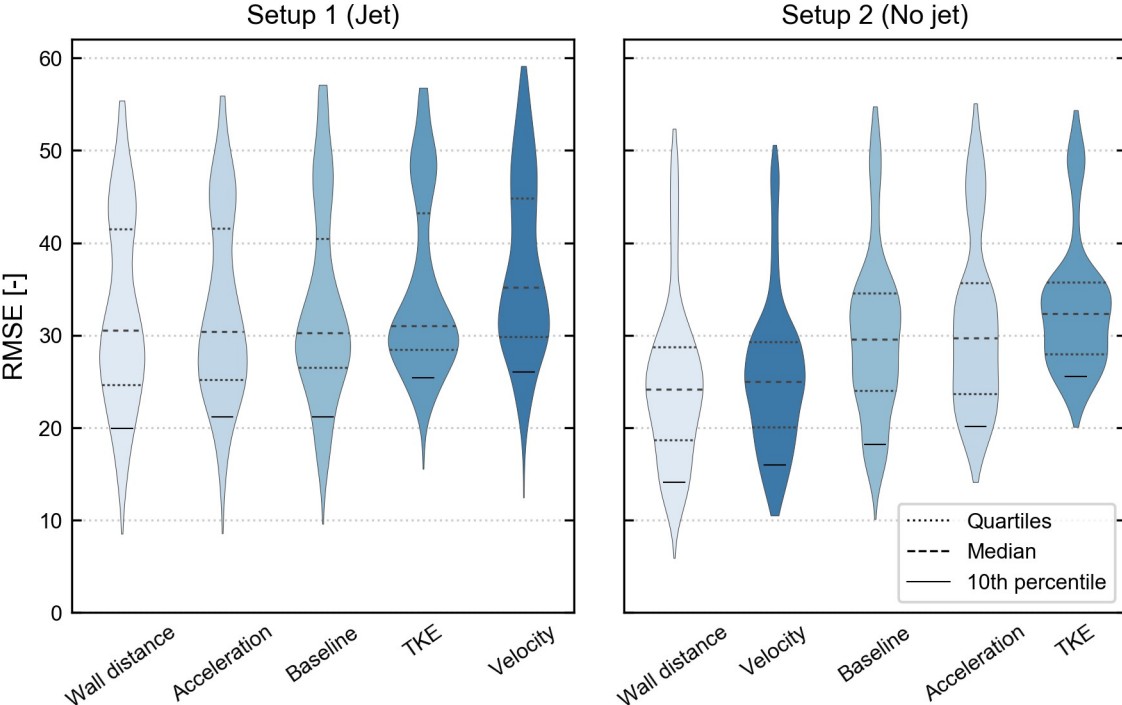

**Fig 5. Violin plots indicating the frequency of root-mean-square error values (RMSE) in both setups.** Ordering the stimulus versions by increasing 10th percentile reveals similarities and significant differences. Results are better if RMSE is closer to zero. The lower violin tip indicates the result of the best parameter set(s) for each stimulus version. Underlying data in S6 Appendix.

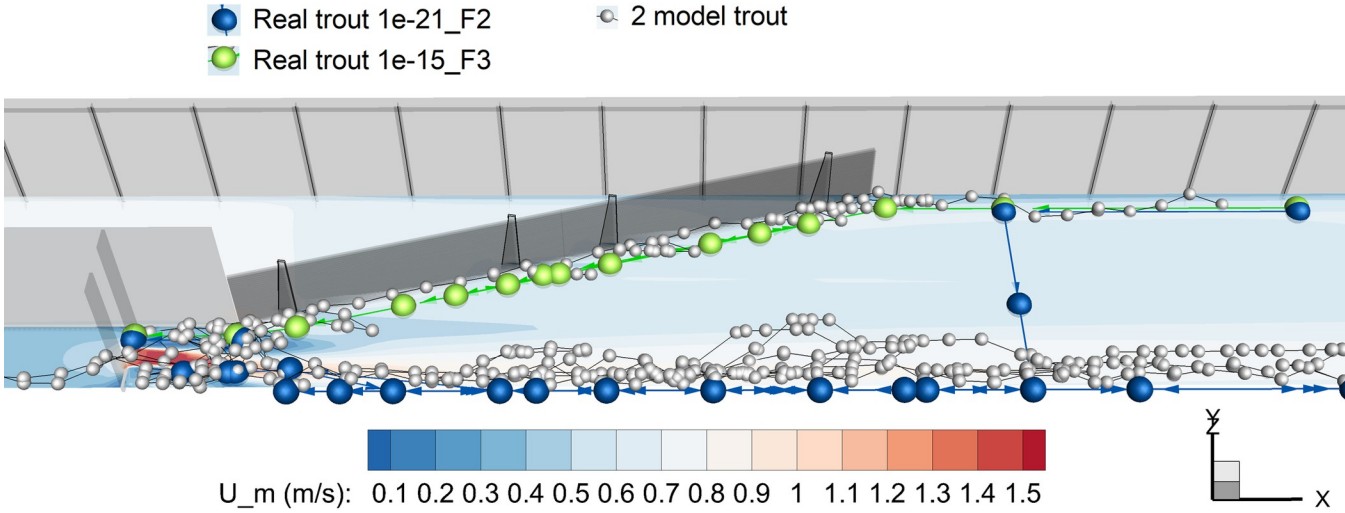

**Fig 6. Two selected real and two selected model trout tracks in setup 1 (with jet).** Each sphere shows one noted position or model time step, respectively. Model results are from the best parameter set of the wall distance stimulus. Plane shows mean velocity, $U_m$, at $z = 0.07$ m above the bottom. Flow direction is from left to right, fish start from right on both flume sides. See S1 and S2 Appendices for observed trout movement data.

and significant differences between the model stimulus versions enable us to develop explanations with respect to the ambient conditions in our two setups.

RMSE rating results for the baseline version ranged from moderate in setup 1 to good in setup 2, indicating that rheotactic orientation and a random component may already explain not only orientation (pattern P1), but also large parts of the remaining observed behavior captured by P2-P5. Positive rheotaxis was expected to be important as both velocity direction and observed movement directions were largely uniform and aligned, fostering this basic behavior [17]. Also, this result confirms that the random component was chosen in the right order to cover effects not modeled explicitly. Still, it was unexpected that the other hydraulic variables commonly associated with behavior, i.e. velocity magnitude, TKE, and acceleration [e.g., 8, 20, 39], were not required for this result. The most apparent explanation that the flow field was not challenging enough to influence trout behavior will be discussed along with the alternative stimulus versions tested.

The velocity version yielded poor results in setup 1 and good results in setup 2. One reason for the failure in setup 1 was that model fish mostly avoided the jet of higher velocity on the right-hand side, while observed fish frequented this zone in both setups (P1). The laboratory observations indicate that the tested trout did not aim to minimize their energy consumption by following low-velocity paths upstream as could be expected [18, 22]. Hence, the good model results in setup 2 are not the result of a real mechanism. They are rather a model effect of the low-velocity zone on the right-hand side, which fostered wall attraction and thus agreement to the observations. Or, put differently, there is no apparent reason why trout would seek lower velocity in setup 2, but would not do so in setup 1 (with larger bulk velocity). In conclusion, our model and laboratory results suggest that velocity was not a relevant stimulus for our trout.

To find an explanation, we used the body length as a proxy for swimming ability and considered the velocity/body length relation. A trout (BL = 0.27 m) holding station experienced a relative flow velocity of about $U_{fish} = 3.0$ BL/s in large parts of the jet in setup 1. At our water temperature, trout can maintain a sustained swimming speed of $U_{fish} = 4.5$ BL/s for up to 200 min according to Ebel's model equation for the rheophilic guild [38]. Thus, in theory, trout

could hold or move upstream while experiencing almost no fatigue [42] for longer than the test duration, indicating that the flow velocity was indeed too low to trigger a reaction. Some support for this explanation comes from two studies, where low-velocity seeking was observed only at much larger relative velocities. Broadly estimated relative velocities were (0.37 m/s)/ (0.07 m/BL) ≈ 5.3 BL/s for Duboulay's rainbowfish [18] and (0.39 m/s)/(0.06 m/BL) ≈ 6.5 BL/ s for longnose dace [39]. In contrast, [22] reported low-velocity seeking already at (0.40 m/s)/ (0.14 m/BL) ≈ 2.9 BL/s for small hatchery brown trout, which is similar to our relative velocity. This discrepancy indicates that a general movement hypothesis has to include more factors than just relative velocity.

Our next test for constant TKE attraction scored poor results in both setups. As both walls and bottom were smooth, they did not provide a distinct TKE stimulus for wall attraction, resulting in large deviations in P1 and the RMSE. These results challenge the statement that TKE is "by far the best stimulus" for fishway models [12] and the hypothesis that a constant turbulence level can be employed for orientation [20]. We did not test attraction towards increasing or decreasing TKE, but would expect similarly poor results for P1 as the TKE field offers a decent stimulus only towards one side (lateral asymmetry), while the fish observed visited both sides.

Like for velocity, we compared TKE levels with literature values to determine if a relation to body length could explain our results. Our setups exhibited levels of TKE = 0.001–0.1 J/kg. Studies which explained behavior using TKE reported higher values for smaller fish: TKE = 0.1–0.3 J/kg for trout, BL = 0.20 m [12] and TKE = 0.015 J/kg for dace, BL = 0.07 m [20]. This qualitative comparison is in agreement with the explanation that the flow field did not cause behavioral reactions as it was not challenging for trout. However, TKE is less studied for orientation than velocity and further research is required.

Results of the acceleration version (good and moderate, respectively) were similar in absolute RMSE terms between setups, but differed widely in their in-setup rank because of better performances of the velocity and baseline versions in setup 2. Finding a theoretical explanation is difficult, as the few existing quantitative studies dealing with acceleration as an orientation stimulus focus on downstream movement [8, 43], which is associated with different responses than upstream movement. As for velocity (but with the better setup swapped), we suspect that a model effect, and not a real mechanism, is responsible for the different ranking: Acceleration amplifies the velocity gradient at the lateral walls and produces a distinct stimulus for model wall attraction in setup 1, but not in setup 2, thus failing to match P1 observations (which are similar in both setups). Taken together, the most probable explanation for our inconsistent results with acceleration, velocity, and TKE is that the live trout did not orient by means of the flow field (except for direction), but by other, non-hydraulic factors. Among these, the most appropriate candidate is non-hydraulic perception of wall distance.

Using the wall distance version, our model performed well in terms of absolute RMSE and significantly better than any alternative version in at least one setup. This result is in line with the assumed limited hydraulic influence on orientation in our conditions. As this model version is based on following a gradient pointing towards the wall, it carries the risk of imposing [6] large residence durations near the walls. This risk is mitigated by two mechanisms for moving away from the wall, namely the random overlay angle and inclined drift. The good RMSE results for this version suggest that these mechanisms are in the right balance and that wall attraction by distance was a real behavior. Potential reasons include guidance, cover, or hydraulic advantages from the walls, but further experiments are required to evaluate their probability.

For the physiological mechanism of wall attraction, multiple explanations are conceivable. To us, visual distance estimation is the best explanation, considering that vision is a

predominant stimulus in fish [16, 25, 44]. Given the clear visibility of the walls, it seems unlikely that another potential far-field stimulus (e.g. acoustic) dominated distance estimation. Distance estimation by means of a hydraulic signature is also unlikely according to our CFD model results: as discussed before, they show that wall influence was limited to some centimeters distance and/or was inconsistent between setups in hydraulic variables (velocity, TKE, acceleration) which are commonly associated with behavioral responses [e.g., 8, 20, 39]. As a study on blind cave fish shows, hydrodynamic imaging by the lateral line close to smooth walls is limited to a small fraction of the body length [45].

Taken together, our results indicate that the rarely considered wall distance can be a relevant stimulus for orientation in fishways and similarly confined migration paths. It requires, of course, ambient conditions which enable its perception, e.g. clear and light water. Further, it may be overruled by hydraulic stimuli if they are relatively larger (in relation to fish swimming ability) than in our conditions.

### Stimulus combinations

In our model, we limited orientation stimuli to the baseline velocity direction and one additional stimulus at a time. This is a simplification of reality, where most complex behavior is unlikely to depend on a single stimulus/sense alone [20, 46], and is more likely to have a polysensory background [47]. While making a model more realistic is not the same as making it more useful, combining stimuli to act on the same behavior is a worthwhile research direction for improving the generality of fish orientation IBMs.

A recent example is the use of calibrated weight factors for three hydraulic stimuli in an upstream orientation IBM for carp [13]. A further step would be to model shifting influence of stimuli depending on the ambient conditions, which was shown to exist in different species [16, 48]. However, adapting our model to use e.g. acceleration in setup 1 and velocity in setup 2 would still yield results not as good as the wall distance version (two times rank 2 vs. two times rank 1). In addition, both stimuli were likely not real stimuli (as discussed above) and hence such a model version would reduce robustness and explanatory power for our experimental conditions. Therefore, we decided not to run our model with stimuli combinations.

### Movement patterns in the laboratory

Transfer of our model also depends on how universal the underlying movement patterns are. The observed tendency to swim close to the walls (pattern P1) matched descriptions of rainbow trout behavior [49] and barbel behavior in a confined model fishway [48]. It was not observed in faster flow and a more narrow flume for brook and brown trout [37]. This discrepancy indicates a potential dependence on the ambient conditions. For our slow, shallow flow, we conclude that P1 is not an untypical behavior.

Swimming at the bottom is reported frequently, e.g. in a natural stream for brown trout [50], and in flumes for rainbow trout [49], barbels [21], brown and brook trout [37], and longnose dace [39]. The water column in our experiment was shallow ($d = 0.60$ m), hence it was unclear whether fish preferred the bottom for orientation, for shelter, or for another reason. Despite possible behavioral or physiological reasons, P2 can be considered a typical behavior in our flume conditions.

For patterns P3 and P4, we did not find matching descriptions in the literature, because they represent a new way of systematically dealing with the striking feature "turns". P5 was also too specific to our setup.

From the literature comparisons, it seems plausible that patterns P1 and P2 would be universal enough to support model transfer to similar flows, e.g. to an altered flume geometry.

Transfer to the real world could be challenging, e.g. due to limited wall distance estimation and/or larger flow velocities.

## General limitations

Finally, we want to point out some general limitations of this work to facilitate critical reception. First, our results are limited in temporal and spatial scale to the estimated fish observation accuracy, which was coarser than the scale of changes in the hydraulic variables within a total distance of 1–2 m downstream of the slot. Thus, potential effects of sudden changes in this area on the behavior could not be incorporated into the model.

Second, we found that different parameter combinations produced similarly good results, which indicates that parameter values are not robust and transfer of the behavioral model to other species and/or hydraulic environments requires recalibration. This does not affect our reproducible testing procedure of complex hypotheses, which is one of the major assets of IBMs [4].

Third, the behavioral model was sensitive to changes in time step width. Main influences changing with the time step include accuracy of flow field perception [9] and the number of orientation and random decisions per time. In our view, even the larger time step tested ($\Delta t$ = 1.0 s) should be sufficiently fine to meet our respective requirements. Hence, this result likely emphasizes sensitivity of the parameter calibration regarding the chosen time step rather than some physical meaning.

Fourth, matching P1 depended, among other effects, on the initial distribution of fish positions to either side. The share of fish starting on the left flume side in the best wall distance version was off the real distributions observed in the flume (33 % vs. 68 % with jet and 33 % vs. 46 % without jet). Producing matching results from differing initial conditions could point to a difference between model and laboratory behavior or it could be a sign of minor relevance of the initial conditions to the model result.

Fifth, our model is currently limited to steady flow fields. Modeling e.g. utilization of transient vortices would require considerable model modification and computational resources [14].

Sixth and last, although swimming in groups (shoaling) was not pronounced in our data set and not modeled, it can influence orientation behavior. Determining its influence would require laboratory experiments with single fish releases.

## Conclusions

Our goal in this study was to better understand upstream fish orientation and navigation in confined space, e.g. in fishways, by means of behavioral rules. Our thoroughly tested IBM enabled representation of orientation choices with an accuracy in the order of degrees. It was able to reproduce five movement patterns of brown trout in two experimental flow fields, which represented a broad range of behavior in a wide flume. The first two patterns–preferring wall and bottom proximity–were very distinct, and should be investigated further to determine their dependency on water width and depth.

The significant advantage of the wall distance stimulus version illustrates that a focus on hydraulic stimuli for predicting fish orientation can be too narrow, especially in hydraulic conditions not challenging for the observed individuals. These can occur e.g. for strong swimmers in multi-species fishways. In such conditions, particularly when the walls are visible, wall distance should be considered more frequently as a stimulus in IBMs, laboratory experiments, and field studies.

The evaluation of observed patterns and key parameters indicates that transfer of our present model is likely limited to similar species and conditions, i.e. brown trout of a given size in a relatively slow, shallow flume flow. This does not affect the IBM's ability to contrast different behavior hypotheses.

In summary, the interdisciplinary IBM technique facilitates testing of behavioral rules using alternative stimuli with flexibility and rigor. It is an important option to approach the many urgent questions of fishway designers.

## Supporting information

**S1 Appendix. Brown trout movement observations, setup 1.** Unpublished movement data from Schütz et al. [28].
(XLSX)

**S2 Appendix. Brown trout movement observations, setup 2.** Unpublished movement data from Schütz et al. [28].
(XLSX)

**S3 Appendix. Hydraulic field data.** ZIP file containing S3a_hydraulic_fields_setup1.csv and S3b_hydraulic_fields_setup2.csv. For coordinate system origin, see Fig 1.
(ZIP)

**S4 Appendix. Behavioral model description.**
(PDF)

**S5 Appendix. Behavioral model code.** The code is written in C++ and Fortran 90. The Fortran code uses the FORTRAN 77 fixed format for historic reasons. https://github.com/baw-de/ELAM-flume
(TXT)

**S6 Appendix. Root mean square errors.** The order is Wall distance (setup 1, setup 2)–Acceleration 1, 2 –Velocity 1, 2 –TKE 1, 2 –Baseline 1, 2.
(CSV)

## Acknowledgments

We thank all persons involved in the realization of the brown trout movement experiments. We thank R. Andrew Goodwin of USACE for providing the Fortran 90 model code used in [8] as a starting point.

## Author Contributions

**Conceptualization:** David C. Gisen, Roman B. Weichert.

**Data curation:** David C. Gisen, Cornelia Schütz.

**Formal analysis:** David C. Gisen.

**Funding acquisition:** David C. Gisen, Roman B. Weichert.

**Investigation:** David C. Gisen.

**Methodology:** David C. Gisen.

**Project administration:** David C. Gisen.

**Resources:** David C. Gisen, Cornelia Schütz.

**Software:** David C. Gisen.

**Supervision:** Roman B. Weichert.

**Validation:** David C. Gisen.

**Visualization:** David C. Gisen.

**Writing – original draft:** David C. Gisen.

**Writing – review & editing:** David C. Gisen, Cornelia Schütz, Roman B. Weichert.

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
