## [Decision Letter · Decision Letter 0]

22 May 2021

PONE-D-21-08612

Development of behavioral rules for upstream orientation of fish in confined space employing wall distance and hydraulic stimuli

PLOS ONE

Dear Dr. Gisen,

Thank you for submitting your manuscript to PLOS ONE. After careful consideration, we feel that it has merit but does not fully meet PLOS ONE’s publication criteria as it currently stands. Therefore, we invite you to submit a revised version of the manuscript that addresses the points raised during the review process.

We look forward to receiving your revised manuscript.

Kind regards,

Atsushi Fujimura

Academic Editor

PLOS ONE

Journal Requirements:

Reviewers' comments:

Reviewer's Responses to Questions

**Comments to the Author**

1. Is the manuscript technically sound, and do the data support the conclusions?

Reviewer #1: Yes

Reviewer #2: Yes

Reviewer #3: Yes

2. Has the statistical analysis been performed appropriately and rigorously? 

Reviewer #1: I Don't Know

Reviewer #2: Yes

Reviewer #3: Yes

3. Have the authors made all data underlying the findings in their manuscript fully available?

Reviewer #1: Yes

Reviewer #2: Yes

Reviewer #3: Yes

4. Is the manuscript presented in an intelligible fashion and written in standard English?

Reviewer #1: No

Reviewer #2: Yes

Reviewer #3: Yes

5. Review Comments to the Author

Reviewer #1: The introduction is poorly organized and confusing. The discussion of hydraulic parameters is insufficient and unclear.

Why would learning effects matter in this scenario?

I don't understand why the fish swimming study was designed the way it was. Was it specifically designed to calibrate/validate the model?

I cannot provide a good review of the movement patterns section (2.2) because it is not something I've studied.

Please provide more information on CFD model setup, calibration and validation.

I would like more explanation for why decisions were made ....e.g. why use averaged/steady results from the CFD model?

I love this stuff and I think its interesting work, I just want more information in the methods that is clearly organized and suggest a full rewrite where you focus on streamlining terminology, writing good paragraphs, and correcting grammar errors. I'm a bit concerned that it is unclear how much of this work is Goodwin's and how much is new work.

Reviewer #2: Nicely thought out work. Please see manuscript PDF where I mention a few places where better word choice will help the reader. More broadly, I think the manuscript could be improved in terms of readability by improving the flow of information. Some detail and terms within the manuscript make it hard to follow at times, so perhaps consider moving some technical pieces/info together (or use more general terms where possible) to allow more portions of the manuscript to flow (read) easier.

Reviewer #3: In this manuscript, a multi-faceted study is used to develop a framework to model fish movement to test different behavioral rules to explain fish orientation and navigation. The study uses fish tracking data from an unpublished laboratory study of brown trout passage through a partitioned flume. Pattern orienting modelling was used to capture 5 distinct movement patterns. The patterns were used as a metric for which to evaluate the performance of the behavioral rules. The authors use the Eulerian-Lagrangian agent method (ELAM) developed by Goodwin et al. (2014) as a foundation for their individual based model (IBM). The authors IBM was used to evaluate three behavioral rules using five different guidance stimuli. The authors found that wall distance guidance stimulus performed best at replicating the movement patterns. Overall, the manuscript was well written and combined with the extensive detail in the appendices (e.g., ODD report) provides a reasonable account of a complex study. Several aspects of the study are not entirely novel. The laboratory study is addressed in a separate manuscript, the CFD analysis is standard practice, and the ELAM provides the general computational infrastructure for the IBM. Therefore, the novelty of this study lies entirely within the behavioral rule and guidance stimulus selection and analysis. While the sensitivity analysis and modelling output are rigorous, the findings are somewhat muted. As detailed later, I found the laboratory study to be incongruous with the movement hypotheses being tested, likely leading to the less explanatory stimulus of wall distance being the best fit. Despite this issue, I feel the manuscript is still an important finding in the field of predictive modelling of fish movement. Due to the number of comments and level of effort required to address, I recommend the manuscript undergo major revisions.

Detailed and line-by-line comments are provided below:

Introduction

1. Line 42 – A reference to Goodwin et al. (2014) would seem appropriate here.

2. Line 48 – While the introduction touches briefly upon aspects of the fuller study, it lacks sufficient detail to understand why certain selections of behaviors and guiding stimuli were chosen. In this instance, citing the IPOS framework to describe what aspects of turbulence are important to fish behaviors would be relevant.

3. What about motivation to move or context specific behaviors? The authors need to address the complexities caused by differences in the internal state of a fish to make different decisions to the same stimuli.

4. The introduction would further benefit from more details on how the current effort extends or differs from previous IBMs. Specifically, it is not clear how the model is differentiated from the ELAM used by Goodwin et al. (2014). The only reference to the ELAM in on Lines 73 and 84 stating the proposed model is “ELAM-type”. I feel the general reader is not going to understand what this means.

5. The authors need to more explicitly state what hypotheses they are testing and why. The preceding paragraphs lists evidence that points to a lot of hydraulic variables that could be influential to fish movement. The authors do not explain why they examine baseline rheotaxis, velocity magnitude, TKE, flow acceleration, and wall distance relative to other choices including turbulent intensity, velocity gradients, eddy sizes, etc.

Methods

6. Line 110 – Where does the “x=9.74” come from? Figure 1 indicates an observation point at x=7.49.

7. Figure 1. The figure caption is the first mention of patterns P1 and P5 without defining them. I suggest moving this statement into the main text after P1 and P5 are defined.

8. Line 109-119 – The methods and scales at which fish movement was tracked is unclear. Where observers tracking movements in real-time on paper as well as noting location relative to the wall or screen or position in a group? How often was the position recorded? The authors state that observations were verified qualitatively with video records, why not use the video to obtain more detailed tracks. Overall, the methods on how tracking was accomplished needs significant more detail.

9. How were the set-ups chosen for the laboratory tests and how does this relate to the central hypotheses being tested?

10. Line 121-147 – How were these patterns chosen and what hypotheses drove these decisions? Their selection seems somewhat random as written. For example, what details informed splitting the channel at a distance of 0.25 m? Does this distance coincide with an observed hydraulic feature or behavior?

11. Line 155 – Omission of the screen bars is not sufficiently detailed in the methods. I understand the reason for not modelling the bars explicitly, but why not model them as a permeable surface to replicate some of the fine-scale turbulence. See Ho et al. (2011).

Ho, J., Coonrod, J., Hanna, L.J., and Mefford, B.W. 2011. Hydrodynamic modelling study of a fish exclusion system for a river diversion. River Res. Applic. 27: 184–192. doi:10.1002/rra.1349.

12. Line 172 – Again, what underlying hypothesis drove the selection of just these 3 hydraulic variables?

13. Line 200 – Was distance to the floor included in the wall distance evaluations?

14. Since the behavioral rules and selection of stimuli are the main contributions of this work, I found the model description provided in the main text to be underwhelming. I generally understand the adherence to the ODD protocol, but this should not sacrifice the completeness of the main text to act as a standalone document.

Results

15. Table 1 – The general magnitude and relation of patterns appears to be nearly identical between set-ups. I would even doubt there is any statistical difference between values. This would indicate one of two possible failures in the experimental design: 1). The modifications to the laboratory set-up did not achieve a discernable change in behavior; or 2). The patterns do not adequately capture the behavioral changes caused by the modification to the set-up. Either way, since the authors do not provide any rationale as to what the set-up change was indented to do makes interpretation difficult.

16. Figure 6 – This figure was very helpful to understand the model results and laboratory data. Additional versions of this figure to compare the modelled movement against observed movement would be beneficial. Illustrations from both set-ups should be included.

Discussion

17. Considering that the observed movement patterns were not largely different between the two set-ups, the conclusion that velocity does not play a role in orientation and navigation is stated too strongly. Based on the available data, it would appear that fish largely exhibited exploratory behavior and followed the walls because the hydraulics did not require any modified behavior. The authors should at least comment that their rule of avoiding high velocities is rather simple. It would not appear to fit the behavioral data either, since passage required movement through an area of elevated velocity in set-up 1. Perhaps a more refined guidance rule surrounding velocity fields is necessary.

18. Line 530-532 – The rationale for not running the model with combinations of stimuli is lacking. I can understand the reason for not including this step in this study, but to dismiss their validity is premature. The authors do not explicitly indicate how multiple stimuli could be integrated into a single model, so I do not understand how they can claim the acceleration and velocity versions could not improve upon the wall distance version without explicitly testing this.

19. Line 589 – The authors repeatedly indicate that wall perception could be accomplished by vision or audition, but it is not entirely clear how or what role audition may play in wall distance perception. I would still expect near field hydrodynamics would override any true acoustical input for most fish. This just strikes me as an odd statement and should be further explained if it remains in the manuscript.

Appendix S4

20. Page 4, Para. 1 – The authors state that the relationship between fatigue and swimming speed is not well understood. I find this statement to be highly inaccurate. Numerous studies are available that examine brown trout (and similar salmonids) swimming performance, that is their swim speed vs. duration curves. The authors also site work by Castro-Santos et al. (2005 and 2013) that shows how swimming fatigue could be modelled and swimming performance data for brown trout.

21. Page 8, Table 3 – How was Ug estimated? Castro-Santos et al. (2013) clearly shows laboratory data on brown trout indicated an optimal ground speed of 5.65 BL/s.

22. Page 10, Section 7.4 – Energy usage is cumulative at the time scale of this study. Once fish expend energy to swim above their sustained swim speed, they are unlikely to recover within 60 mins. However, since the authors assume no fish became fully exhausted from swimming in the flume, their handling of fatigue as a behavioral metric is understandable. This caveat could be clearer in the main text.

23. Page 6, Section 4.7 – The authors should comment on whether the random effects encompass stochasticity in animal behavior and the hydraulic stochasticity (i.e., the real flow field is not static).

6. PLOS authors have the option to publish the peer review history of their article (what does this mean?). If published, this will include your full peer review and any attached files.

Reviewer #1: No

Reviewer #2: **Yes: **R. Andrew Goodwin

Reviewer #3: No

---

## [Author Response · Author response to Decision Letter 0]

5 Jul 2021

PLEASE SEE BELOW FOR ORIGINAL TABULAR FORM

Response to reviewers

First, we would like to thank the reviewers for their insightful comments, which helped us to improve the manuscript. Our particular responses can be found below. Generally, we tried to further streamline terms, correct errors, and to improve readability throughout the whole manuscript.

We note that the manuscript does not yet meet the style requirements of PLOS ONE, but decided to await the provisional Editorial Accept decision (per https://journals.plos.org/plosone/s/getting-started) to avoid confusion during review e.g. by changing the citation style.

Reviewer #1

Comment Addressed? How addressed or why not? Line (original)

1. The introduction is poorly organized and confusing. The discussion of hydraulic parameters is insufficient and unclear. Yes We’ve rewritten large parts of the introduction, focusing on readability and the motivation for our study. We’ve also deleted information on stimuli that went too much into detail or - where helpful - moved them to the model description to provide a rationale on why each stimulus was chosen. 

2. Why would learning effects matter in this scenario? No Fish already familiar with the flume geometry and hydraulic field could deviate in their behavior unpredictably, e.g. navigate straight towards the upstream end or don’t move at all. Excluding such effects by testing every individual just once makes results more reproducible. 110

3. I don't understand why the fish swimming study was designed the way it was. Was it specifically designed to calibrate/validate the model? Yes (See also comments #9 and #15 of reviewer #3.)

The previous, unrelated study was primarily designed to investigate the influence of auxiliary water velocity coming from the screen on passage success and delay of five species. 

Conditions were chosen to create hydraulic conditions typical for a large multi-species fishway in Germany. It is now published in Ecological Engineering, where more information is available: https://doi.org/10.1016/j.ecoleng.2021.106257

We used the both setups to establish an application range for our model, see also comments #9 and #15 of reviewer #3.

We’ve added information on design choice and the word “unrelated” in section 2.1 for clarification. 100

4. I cannot provide a good review of the movement patterns section (2.2) because it is not something I've studied. - - -

5. Please provide more information on CFD model setup, calibration and validation. No We feel the information provided on the model setup is sufficient for reproduction. Calibration in the classical sense was not performed, as boundary conditions are fully determined by the laboratory flume. Mesh, turbulence model, and screen negligence were validated using ADV probe data (line 362f). Comprehensive detail information can be found in (Gisen 2018, p. 33ff.). 153ff

6. I would like more explanation for why decisions were made ....e.g. why use averaged/steady results from the CFD model? Yes Using transient CFD results is a promising direction to capture behavior currently modelled stochastically. However, interpolating thousands of 3D fish positions on 7200 flow field snapshots (dt = 0.5 s for 1 h) is currently far beyond our simulation capacity, even using a high-performance cluster. We’ve added a brief mention to the limitations section 4.4. 

7. I love this stuff and I think its interesting work, I just want more information in the methods that is clearly organized and suggest a full rewrite where you focus on streamlining terminology, writing good paragraphs, and correcting grammar errors. Yes We’ve added information to the methods section and tried to streamline terminology in the whole manuscript. We considered to change the order of subsections in section 2.5, but refrained from doing so, as there are too many dependencies in both ways, e.g. between RMSE, RMSE(model-run), and parameter set evaluation. We rely on figure 3 to explain these dependencies. 

8. I'm a bit concerned that it is unclear how much of this work is Goodwin's and how much is new work. Yes We’ve tried to clarify in the introduction that Goodwin (2014) worked on downstream migration on dam-scale, while we work on upstream migration on fishway-scale, which required e.g. a completely new behavioral model. The CFD model was also replaced, of course, and the software framework was heavily modified (section 2.4.1). 

Reviewer #2

Reviewer #2: “Nicely thought out work. Please see manuscript PDF where I mention a few places where better word choice will help the reader. More broadly, I think the manuscript could be improved in terms of readability by improving the flow of information. Some detail and terms within the manuscript make it hard to follow at times, so perhaps consider moving some technical pieces/info together (or use more general terms where possible) to allow more portions of the manuscript to flow (read) easier.”

Comment Addressed? How addressed or why not? Line (original)

1. Unclear, please restate another way. Yes Rephrased 50

2. Unclear. Yes Replaced “like for P1” with “using Eq. 1” 136

3. Unclear, please restate another way. Yes Rephrased 321

4. Unclear, please restate another way. Yes Rephrased 322

5. underlying ? Yes Deleted “a” 537

6. misplaced "e.g." ? No Other influences than width and depth are conceivable, such as smooth boundaries and illumination 584

7. demanding enough for No From our perception “enough” would suggest a need to create more demanding hydraulic conditions, which was not our intention in this study (see reviewer #1, comment #3). 586f.

Reviewer #3

Reviewer #3: “In this manuscript, a multi-faceted study is used to develop a framework to model fish movement to test different behavioral rules to explain fish orientation and navigation. The study uses fish tracking data from an unpublished laboratory study of brown trout passage through a partitioned flume. Pattern orienting modelling was used to capture 5 distinct movement patterns. The patterns were used as a metric for which to evaluate the performance of the behavioral rules. The authors use the Eulerian-Lagrangian agent method (ELAM) developed by Goodwin et al. (2014) as a foundation for their individual based model (IBM). The authors IBM was used to evaluate three behavioral rules using five different guidance stimuli. The authors found that wall distance guidance stimulus performed best at replicating the movement patterns. 

Overall, the manuscript was well written and combined with the extensive detail in the appendices (e.g., ODD report) provides a reasonable account of a complex study. Several aspects of the study are not entirely novel. The laboratory study is addressed in a separate manuscript, the CFD analysis is standard practice, and the ELAM provides the general computational infrastructure for the IBM. Therefore, the novelty of this study lies entirely within the behavioral rule and guidance stimulus selection and analysis. 

While the sensitivity analysis and modelling output are rigorous, the findings are somewhat muted. As detailed later, I found the laboratory study to be incongruous with the movement hypotheses being tested, likely leading to the less explanatory stimulus of wall distance being the best fit. Despite this issue, I feel the manuscript is still an important finding in the field of predictive modelling of fish movement. Due to the number of comments and level of effort required to address, I recommend the manuscript undergo major revisions.”

Comment Addressed? How addressed or why not? Line (original)

Introduction 

1. Line 42 – A reference to Goodwin et al. (2014) would seem appropriate here. Yes The reference Silva et al. 2018 serves to underline the “high research priority” of behavioral rules. We did not intend to reference general studies using behavioral rules here, elsewise we would have to list much more than just Goodwin’s. We’ve tried to make this clearer by using a direct quote of Silva et al.

2. Line 48 – While the introduction touches briefly upon aspects of the fuller study, it lacks sufficient detail to understand why certain selections of behaviors and guiding stimuli were chosen. In this instance, citing the IPOS framework to describe what aspects of turbulence are important to fish behaviors would be relevant. Yes For choice of behaviors and stimuli see response to comment #1 of reviewer #1.

We’ve added a reference to IPOS (DOI:10.1002/rra.1584) for general understanding. However, for our IBM study (and others we are aware of), periodicity, orientation, and scale are still too detailed (see comment #5 below and new reference Roth et al. 2021). 

3. What about motivation to move or context specific behaviors? The authors need to address the complexities caused by differences in the internal state of a fish to make different decisions to the same stimuli. Yes This is an important aspect and we’ve added a corresponding paragraph in the introduction. As we focused on orientation and fishway design, we modeled internal stimuli only rather basically (using proxies for motivation and fatigue). We now discuss this in the model description (2.4). 

4. The introduction would further benefit from more details on how the current effort extends or differs from previous IBMs. Specifically, it is not clear how the model is differentiated from the ELAM used by Goodwin et al. (2014). The only reference to the ELAM in on Lines 73 and 84 stating the proposed model is “ELAM-type”. I feel the general reader is not going to understand what this means. Yes See response to comment #8 of reviewer #1.

We also point more directly to lacks of previous upstream IBMs (testing, resting behavior). We’ve removed the term “ELAM”, as it is not required to understand our work. 

5. The authors need to more explicitly state what hypotheses they are testing and why. The preceding paragraphs lists evidence that points to a lot of hydraulic variables that could be influential to fish movement. The authors do not explain why they examine baseline rheotaxis, velocity magnitude, TKE, flow acceleration, and wall distance relative to other choices including turbulent intensity, velocity gradients, eddy sizes, etc. Yes See response to comment 1 of reviewer #1.

We’ve added a justification for each stimulus tested in the behavioral model description (2.4) and a summary in the introduction. Generally, we preferred the more comprehensive variable over specialized variables. E.g. all three turbulence intensity components are contained in the TKE (see e.g. IPOS paper, p. 432).

The velocity gradient tensor is contained in the acceleration metric. 

Strain rate and rotation are contained in the velocity gradient tensor. 

Reynolds shear stresses could be relevant to future work. 

Eddy size, as well as eddy orientation, is an interesting, but advanced measure left for future work (see response #2 above). It is partly covered in the rheotaxis stimulus. 

Methods 

6. Line 110 – Where does the “x=9.74” come from? Figure 1 indicates an observation point at x=7.49. Yes At x=9.74 m, there was an external pole on the flume used as mark for quick orientation in the original study. We’ve removed this information, as it does not affect this manuscript. 

7. Figure 1. The figure caption is the first mention of patterns P1 and P5 without defining them. I suggest moving this statement into the main text after P1 and P5 are defined. Yes We’ve inserted into the caption a reference to section 2.2 where the patterns are defined. If the caption was changed per your suggestion, readers could be confused about the meaning of the gray areas and line D. 

8. Line 109-119 – The methods and scales at which fish movement was tracked is unclear. Where observers tracking movements in real-time on paper as well as noting location relative to the wall or screen or position in a group? How often was the position recorded? The authors state that observations were verified qualitatively with video records, why not use the video to obtain more detailed tracks. Overall, the methods on how tracking was accomplished needs significant more detail. Yes We’ve added this lacking information to the methods (real-time recording on paper; positions were recorded on every notable change; definitions of changes in the three dimensions).

We’ve also added to methods and appendix S1 the information that x_value and y_value were computed in postprocessing.

A manual video analysis of fish tracks would have been too time consuming and (because cameras were only positioned laterally) would not have been reliable for lateral movements; we also worked on an automated 3D-fishtracking of the videos (along 11 overlapping cameras) but with three fish swimming simultaneously this is demanding and proved to be less efficient than the manual records for this project. 

9. How were the set-ups chosen for the laboratory tests and how does this relate to the central hypotheses being tested? Yes (See also response to comment #15 of reviewer #3 and comment #3 of reviewer #1.)

The previous study (now available Schütz et al. 2021, https://doi.org/10.1016/j.ecoleng.2021.106257) was not related to our study. It was designed to investigate the influence of auxiliary water velocity coming from the screen on behavior of five species.

We’ve added the word “unrelated” in section 2.1 for clarification.

We chose two hydraulically different setups (information added) from this study to test the application range of our model/hypotheses. We did not expect the behavior to be this similar (e.g., Kerr 2016 reported low-energy seeking for trout at smaller U=0.40 m/s). 

We still found the result interesting, as it expands the common focus of fishways designers on hydraulic stimuli towards vision under certain conditions (possibly low relative flow velocity and visible boundaries).

10. Line 121-147 – How were these patterns chosen and what hypotheses drove these decisions? Their selection seems somewhat random as written. For example, what details informed splitting the channel at a distance of 0.25 m? Does this distance coincide with an observed hydraulic feature or behavior? Yes Added explanation: To capture the most striking spatial behaviors observed (wall and bottom proximity and turns). 0.25 m was chosen arbitrarily to define wall proximity. 

11. Line 155 – Omission of the screen bars is not sufficiently detailed in the methods. I understand the reason for not modelling the bars explicitly, but why not model them as a permeable surface to replicate some of the fine-scale turbulence. See Ho et al. (2011).

Ho, J., Coonrod, J., Hanna, L.J., and Mefford, B.W. 2011. Hydrodynamic modelling study of a fish exclusion system for a river diversion. River Res. Applic. 27: 184–192. doi:10.1002/rra.1349. No While it is possible to model the screen with bars as a permeable baffle, we avoided the effort of implementation and calibration, as we judged the flow field to be sufficiently close to ADV measurements with respect to our behavioral data accuracy. See also reference Gisen 2018, p. 33ff., for detailed comparisons.

Regarding fine-scale turbulence, we would expect some kind of damping effect from a permeable baffle, which would require extra attention. 155

12. Line 172 – Again, what underlying hypothesis drove the selection of just these 3 hydraulic variables? Yes See response to comment #1 of reviewer #1. 

13. Line 200 – Was distance to the floor included in the wall distance evaluations? No No, it wasn’t, as the vertical behavior was treated as a separate behavioral rule, which relied on the vertical coordinate (~ pressure) as a stimulus. 

14. Since the behavioral rules and selection of stimuli are the main contributions of this work, I found the model description provided in the main text to be underwhelming. I generally understand the adherence to the ODD protocol, but this should not sacrifice the completeness of the main text to act as a standalone document. Yes We’ve added information on motivation and fatigue, the basic internal states of the model and added a reference (Castro-Santos 2004) to substantiate the importance of repeated passage attempts in the model description. 

Results 

15. Table 1 – The general magnitude and relation of patterns appears to be nearly identical between set-ups. I would even doubt there is any statistical difference between values. This would indicate one of two possible failures in the experimental design: 1). The modifications to the laboratory set-up did not achieve a discernable change in behavior; or 2). The patterns do not adequately capture the behavioral changes caused by the modification to the set-up. Either way, since the authors do not provide any rationale as to what the set-up change was indented to do makes interpretation difficult. No (see also response #9)

We agree that our trout behavior as measured by the patterns is very similar between setups. This would be your case (1).

However, the previous, unrelated study (Schütz et al. 2021) did not aim to produce different behavior, but was intentionally limited to a typical maximum slot velocity in a multi-species fishway (1.5 m/s). We’ve added this information in section 2.1.

This limitation coincides with the goal of the present study to provide behavioral rules for fishway design. Hence, the only goal of using two hydraulically different setups was to quantify the (minimum) application range of our model. Finding the maximum application range would be interesting, but less relevant to design of multi-species fishways, which also need to address species with lesser swimming abilities than trout. 

16. Figure 6 – This figure was very helpful to understand the model results and laboratory data. Additional versions of this figure to compare the modelled movement against observed movement would be beneficial. Illustrations from both set-ups should be included. No This figure was intended primarily to foster qualitative understanding. It has little explanatory power, as it depicts only two selected observed and two selected model tracks. Showing more tracks like this would not add quantitative information. Also, a figure showing results for setup 2 would add little additional information (as behavior was not much different). Hence, we would prefer to leave the figure unchanged. 

Discussion 

17. Considering that the observed movement patterns were not largely different between the two set-ups, the conclusion that velocity does not play a role in orientation and navigation is stated too strongly. Based on the available data, it would appear that fish largely exhibited exploratory behavior and followed the walls because the hydraulics did not require any modified behavior. The authors should at least comment that their rule of avoiding high velocities is rather simple. It would not appear to fit the behavioral data either, since passage required movement through an area of elevated velocity in set-up 1. Perhaps a more refined guidance rule surrounding velocity fields is necessary. No By no means we intend to state that velocity does not play are role in orientation, generally. However, we found that in our conditions, which are typical for large multi-species fishways built in Germany, brown trout did not use velocity for orientation (probably because it was not demanding enough). We’ve added explanations to the discussion to clarify this distinction.

Reviewer #3 is right in the point that orientation towards reduced velocity hinders passage through the jet region and slot in setup 1. It is not impossible, however, because the motivation/fatigue driver for movement is independent of the orientation stimulus and there is also a random component to orientation. Further, it cannot be the single cause for the poor performance of the velocity stimulus version in setup 1, as pattern P5 (which measures slot passage) only contributes 20% to the RMSE metric (section 2.5.1).

Still, more refined behavioral rules allowing adaptation to changing flow fields are definitely worthwhile for future work, as discussed in section 4.2. 

18. Line 530-532 – The rationale for not running the model with combinations of stimuli is lacking. I can understand the reason for not including this step in this study, but to dismiss their validity is premature. The authors do not explicitly indicate how multiple stimuli could be integrated into a single model, so I do not understand how they can claim the acceleration and velocity versions could not improve upon the wall distance version without explicitly testing this. Yes Our statement was based on comparing the 10th percentile/rank results of the stimulus versions under the assumption of using only one stimulus per setup. It is true that we did not test using multiple stimuli within one setup. However, we argue that velocity and acceleration are not relevant for brown trout orientation in our flume conditions (see comment and response #17), therefore we would expect similar results from a model using both stimuli in the same setup.

We rephrased the relevant lines to clarify this point. 

19. Line 589 – The authors repeatedly indicate that wall perception could be accomplished by vision or audition, but it is not entirely clear how or what role audition may play in wall distance perception. I would still expect near field hydrodynamics would override any true acoustical input for most fish. This just strikes me as an odd statement and should be further explained if it remains in the manuscript. Yes We agree that this is a hypothetic stimulus on the scales investigated here and removed the word “acoustic” in the abstract and conclusions to make that clearer. 

Appendix S4 

20. Page 4, Para. 1 – The authors state that the relationship between fatigue and swimming speed is not well understood. I find this statement to be highly inaccurate. Numerous studies are available that examine brown trout (and similar salmonids) swimming performance, that is their swim speed vs. duration curves. The authors also site work by Castro-Santos et al. (2005 and 2013) that shows how swimming fatigue could be modelled and swimming performance data for brown trout. Yes Agreed, we deleted this statement. 

21. Page 8, Table 3 – How was Ug estimated? Castro-Santos et al. (2013) clearly shows laboratory data on brown trout indicated an optimal ground speed of 5.65 BL/s. Yes We estimated the value based on Castro-Santos’ 2005 result for striped bass (2.56 BL/s) as stated. We agree that their 2013 paper is a better reference as it deals directly with brown trout. However, note that their value of 5.65 BL/s is a theoretical value for prolonged swim speed and most individuals chose to swim “well below” it (p. 287). Hence, our estimation seems to be in a realistic order. We changed the reference in Table 3, footnote 2, as well as the rationale in section 7.5 accordingly. 

22. Page 10, Section 7.4 – Energy usage is cumulative at the time scale of this study. Once fish expend energy to swim above their sustained swim speed, they are unlikely to recover within 60 mins. However, since the authors assume no fish became fully exhausted from swimming in the flume, their handling of fatigue as a behavioral metric is understandable. This caveat could be clearer in the main text. Yes We did not intend to model fatigue and recovery processes in detail, but just enough to drive model movement as measured by the patterns to enable orientation analyses.

We’ve mentioned the caveat in section 2.4, and included references to Castro-Santos et al. 2013 and Ebel 2014. 

23. Page 6, Section 4.7 – The authors should comment on whether the random effects encompass stochasticity in animal behavior and the hydraulic stochasticity (i.e., the real flow field is not static). Yes We’ve added the explanation that our behavioral model’s stochasticity also covers transient flow field effects on the behavior. S4 section 4.7

---

## [Decision Letter · Decision Letter 1]

22 Oct 2021

PONE-D-21-08612R1Development of behavioral rules for upstream orientation of fish in confined spacePLOS ONE

Dear Dr. Gisen,

Thank you for submitting your manuscript to PLOS ONE. After careful consideration, we feel that it has merit but does not fully meet PLOS ONE’s publication criteria as it currently stands. Therefore, we invite you to submit a revised version of the manuscript that addresses the points raised during the review process.

We look forward to receiving your revised manuscript.

Kind regards,

Atsushi Fujimura

Academic Editor

PLOS ONE

Journal Requirements:

Reviewers' comments:

Reviewer's Responses to Questions

**Comments to the Author**

1. If the authors have adequately addressed your comments raised in a previous round of review and you feel that this manuscript is now acceptable for publication, you may indicate that here to bypass the “Comments to the Author” section, enter your conflict of interest statement in the “Confidential to Editor” section, and submit your "Accept" recommendation.

Reviewer #2: (No Response)

Reviewer #3: All comments have been addressed

2. Is the manuscript technically sound, and do the data support the conclusions?

Reviewer #2: Yes

Reviewer #3: Partly

3. Has the statistical analysis been performed appropriately and rigorously? 

Reviewer #2: Yes

Reviewer #3: Yes

4. Have the authors made all data underlying the findings in their manuscript fully available?

Reviewer #2: Yes

Reviewer #3: Yes

5. Is the manuscript presented in an intelligible fashion and written in standard English?

Reviewer #2: Yes

Reviewer #3: Yes

6. Review Comments to the Author

Reviewer #2: The manuscript is much improved. I have, again, made comments and suggestions (annotations) directly into the manuscript PDF itself, attached as part of this review. There are still a few areas where the authors should add clarity in the narration of their results, especially related to parameter testing. In short, there is a lot of nuance in the parameter analyses and I found it difficult at times to keep track of all the different nuances moving from one paragraph to another. Perhaps consider reminding the reader of some of the basics (setup) which is then elaborated in great detail/nuance within the paragraph.

Reviewer #3: In general, the authors addressed most of the reviewer comments well. Of greatest benefit is the additional discussion on the model limitations. The introduction follows a more logical path and the justification of selected behaviors is much improved. While I still find the experimental setups and movement tracking methods to be less than ideal, I find the modelling results to be worthwhile additions to the body of literature. Specific comments are noted below with the corresponding line number.

Line 87 – I feel that a major contribution of this study is the use of movement parameters to evaluate model performance in a more objective manner than qualitative trajectory comparisons. The authors seem to be hinting at that with this statement, but further details should be added here. Regardless, a paragraph needs to be more than a single sentence.

Line 90 – “For the first time” is an unnecessary detail that exaggerates the novelty of the study.

Line 141 – From the supplementary data, it is clear that observations of trout swimming positions were relatively coarse (> 5 sec) especially in comparison to the output of the modelled positions (0.5 sec). Based on the methods provided, it is unclear how the authors dealt with the difference in temporal resolution for deriving the movement parameters. I suspect this could influence the RMSE values especially with the relatively small spatial region covered by the P1 left and right zones.

Line 196 – Delete “e.g.”

Line 296 – Insert “an” before “evaluation metric”.

Line 420-421 – It is interesting that time step selection had such an influence on model performance, and that is varied between setup, especially for static model environment. The authors should expand on this finding in the discussion.

Line 445 – Insert “in” before “accordance”.

Line 449 – 496 – The method used by Zielinski et al. (2018) only enforced path selection based on energy conservation when fish swam at prolonged and burst swimming speeds, which does not appear to be the case in either setup in this study.

Line 565 – 578 – I think a critical omission in this discussion in the findings of Goodwin et al. (2014) which found the best fitting behavioral model integrated 4 separate behaviors. Without performing runs with multiple behaviors possible, the authors seem to over reach with their dismissal of hydraulic cues influencing brown trout movement. While I agree that hydraulically mediated behaviors are unlikely to be observed in setup 2, certainly as trout near the vertical slot in setup 1 movement should be more complex than simple wall following. I also agree that simplified models, in general, provide the greatest explanatory power and exportability, but this alone should not prevent exploration into more complex models.

7. PLOS authors have the option to publish the peer review history of their article (what does this mean?). If published, this will include your full peer review and any attached files.

Reviewer #2: **Yes: **R. Andrew Goodwin

Reviewer #3: No

---

## [Author Response · Author response to Decision Letter 1]

10 Dec 2021

See tabular document attached (Response to reviewers.docx)

---

## [Editor Report · Decision Letter 2]

2 Feb 2022

Development of behavioral rules for upstream orientation of fish in confined space

PONE-D-21-08612R2

Dear Dr. Gisen,

We’re pleased to inform you that your manuscript has been judged scientifically suitable for publication and will be formally accepted for publication once it meets all outstanding technical requirements.

Kind regards,

Atsushi Fujimura

Academic Editor

PLOS ONE

---

## [Editor Report · Acceptance letter]

4 Feb 2022

PONE-D-21-08612R2 

Development of behavioral rules for upstream orientation of fish in confined space 

Dear Dr. Gisen:

I'm pleased to inform you that your manuscript has been deemed suitable for publication in PLOS ONE. Congratulations! Your manuscript is now with our production department. 

Kind regards, 

on behalf of

Dr. Atsushi Fujimura 

Academic Editor

PLOS ONE